# What Is Wrong with *Frankenia nodiflora* Lam. (*Frankeniaceae*)? New Insights into the South African Sea-Heaths

**DOI:** 10.3390/plants12142630

**Published:** 2023-07-13

**Authors:** Manuel B. Crespo, María Ángeles Alonso, Mario Martínez-Azorín, José Luis Villar, Ladislav Mucina

**Affiliations:** 1Departamento de Ciencias Ambientales y Recursos Naturales (dCARN), Universidad de Alicante, P.O. Box 99, ES-03080 Alicante, Spain; ma.alonso@ua.es (M.Á.A.); mmartinez@ua.es (M.M.-A.); jose.villar@ua.es (J.L.V.); 2Iluka Chair in Vegetation Science and Biogeography, Harry Butler Institute, Murdoch University, 90 South Street, Murdoch, Perth, WA 6150, Australia; ladislav.mucina@murdoch.edu.au; 3Department of Geography and Environmental Studies, Stellenbosch University, Private Bag X1, Matieland, Stellenbosch 7602, South Africa

**Keywords:** *Frankenia*, *Frankeniaceae*, nomenclature, ITS phylogeny, plant endemics, plant morphology, southern African flora, taxonomy

## Abstract

The taxonomic identity and phylogenetic relationships of several southern African perennial taxa related to *Frankenia repens* are discussed. In particular, *F. nodiflora* Lam., a misunderstood species described from the Cape region and synonymised to *F. pulverulenta*, is restored for plants endemic to salt-pans and riverbeds in the coastal lowlands across the Cape Flats (Western Cape province, South Africa). Further, a revision of morphologically close plants, usually identified as *F. pulverulenta* or *F. repens*, also occurring in similar saline ecosystems of the inland western South Africa revealed the existence of two distinct new entities not matching any described taxa of the genus. Molecular analyses of nuclear ribosomal (ITS1-5.8S-ITS2 region) DNA sequence data together with morphological divergence allow recognition of those taxa at species rank, within an independent lineage close to *F. repens*. In consequence, two new sea-heath species are described in the so-called “*F. repens* group”: *F. nummularia* from the Nama-Karoo Biome (Western Cape and Northern Cape provinces), and *F. anneliseae* from the Succulent Karoo Biome (Northern Cape province). Full morphological description and type designation are reported for each accepted species as well as data on ecology, habitat, distribution, and taxonomic relationships to other close relatives are given. Further, an identification key is presented to facilitate recognition of the southern African taxa of *Frankenia*.

## 1. Introduction

*Frankeniaceae* Desv., *nom. cons.*, is a family of mostly halophytic herbs and shrubs, which has been usually accepted to include two to five genera [1,2,3,4]. However, available molecular phylogenies [5,6,7] recovered members of all those genera embedded in a single clade among species of *Frankenia* L., and therefore the latter is currently accepted to be the only genus in the family.

So treated, *Frankenia* s.l. includes about 80–90 species, mostly occurring in saline, gypsum- and carbonate-rich substrates in Mediterranean-type regions around the world, but with ca. 40 species found in Australia [8,9]. Members of that genus, the so-called “sea-heaths”, often occur in coastal and inland saltmarshes, commonly experiencing semi-arid to arid climates, usually having prevalent winter rainfall [10].

Four species are currently recognised as occurring in southern Africa [10,11]. Three of them, *Frankenia repens* (P.J. Bergius) Fourc., *F. pomonensis* Pohnert, and *F. fruticosa* J.C. Manning and Helme, are woody perennial or suffruticose endemics. The two former taxa occur in salt marshes of South Africa and/or Namibia while the latter is a very rare dwarf shrub limited exclusively to quartz outcrops of southern Knersvlakte in Namaqualand (South Africa). The fourth species, *F. pulverulenta* L., is a subcosmopolitan annual herb commonly linked to human disturbance, and it is also widespread in most of South Africa.

Recent taxonomic research conducted in the southern part of Africa revealed the existence of peculiar suffruticose plants occurring on saline soils of the Northern Cape and Western Cape provinces (South Africa), clearly differing from other species of the genus. These plants were usually identified as *Frankenia capitata* Webb and Berth. or *F. repens* (incl. *F. kreibsii* Cham. and Schltdl.), but more often as *F. pulverulenta* L. (subsp. *pulverulenta*), a name having *F. nodiflora* Lam. in synonymy [4,12] (https://powo.science.kew.org/taxon/urn:lsid:ipni.org:names:77229962-1; accessed on 3 April 2023). However, Lamarck [13] described his *F. nodiflora* based on material gathered in an undetermined place in the lowlands surrounding Cape Town (South Africa), and it was said to be distinct from the Linnaean species. In fact, a close examination of Lamarck’s original material reveals a unique combination of morphological characters missing in any of the known South African taxa of the genus. Furthermore, plants akin to *F. nodiflora* occurring in the inland dry areas of Namaqualand and Karoo (western South Africa) also exhibit morphological peculiarities not fitting with the latter, which allow easy recognition when compared to other South African congeners.

In the present contribution, the name *Frankenia nodiflora* is therefore restored for a misunderstood species endemic to the Cape Town lowlands classified in the Fynbos (F) biome. In addition, two new related species are described from the Nama-Karoo (NK) and Succulent-Karoo (SK) biomes [14] in western South Africa. These are named here: *F. nummularia* and *F. anneliseae*. Morphological, distributional, ecological, and molecular–phylogenetic data are reported supporting their recognition at a specific rank. Types are also designated for the concerned taxa, and a tentative taxonomic key is presented for the identification of the known southern African species of *Frankenia*.

## 2. Materials and Methods

### 2.1. Morphological and Habitat Studies

Detailed morphological studies were undertaken using an OLYMPUS SZX7 binocular on both living plants from wild populations and dried herbarium specimens sourced from the herbaria ABH, BM, BOL, G-DC, HBG, K, LINN, M, MA, MW, NBG, P, PRE, PRU, SBT, W, and WU (acronyms according to Thiers [15]). Barcode numbers were placed after the corresponding herbarium acronym when available. Digital images of *Frankenia* from iNaturalist (https://www.inaturalist.org/observations/?place_id=any&taxon_id=58170; accessed on 3 April 2023) were also checked and considered to fulfil distributions of the concerned species.

When available, at least 10 mature seeds of several individuals from at least three populations of the studied species (Table 1) were observed in detail for morphological characterisation. Seeds were taken from both living plants and herbarium sheets, and when possible, they were rehydrated for more accurate observations. Scanning Electron Microscope (SEM) micrographs of seeds were taken with a JEOL (Tokyo, Japan), JSM-IT500HR operating at 15 kV. No special treatment of the material was required prior to observation. Samples were glued directly on metallic stubs and then coated with 10 nm platinum in a QUORUM Q150T ES Plus sputter coater. The ImageJ software V1.8.0 [16] was used for measurements on SEM micrographs.

Authors of the taxa cited in the text followed IPNI [17]. Nomenclatural issues followed Turland et al. [18]. Bioclimate, bioregion, and vegetation classification agreed with Mucina and Rutherford [14].

### 2.2. Geographic Coding of the Vouchers and Observations

Orthography of geographical names and grid-number system accorded with Leistner and Morris [19], and the grid–number system followed the National Geospatial Information (http://www.ngi.gov.za/indexphp/what-we-do/maps-and-geospatial-information/41-sa-mapsheet-referencing; accessed on 3 April 2023). The geographic coding of the localities of collected vouchers (e.g., 3318CD or −CD, always linked to 4-digit code) represented the centre point for a 15′ latitude × 15′ longitude sub-tile as defined by the South African topographical map sheet referencing system (National Geospatial Information 2020). This system is also known as ‘quarter degree square’ (QDS) grid since each topographic map sheet is considered a tile (coded, for instance, as 3318) divided into four sub-tiles (coded, for instance, as −CD).

### 2.3. Molecular Analyses

Herbarium vouchers and silica gel-dried material were used for total DNA extraction employing a modified 2 × cetyltrimethylammonium bromide (CTAB) protocol [20]. When sampling from herbarium material was not permitted, only silica-gel dried material from wild populations (one sample per population) of each taxon was utilised if available since addition of new samples from the same populations did not modify the phylogenetic trees. In the case of *F. nodiflora*, unfortunately, we were only able to get a reduced amount of DNA from a small sample not properly preserved, which did not amplify correctly, and the obtained sequences were unusable. Therefore, it was not included in the phylogenetic analyses. Total DNA was purified using MOBIO minicolumns and kept in 0.1 × TE buffer [10 mM Tris-HCl, 1 mM ethylenediaminetetraacetic acid (EDTA), pH 8.0]. The internal transcribed spacer –ITS– region (ITS1 spacer, 5.8S gene, ITS2 spacer) of nuclear ribosomal DNA (nrDNA) was amplified using the ITS5 and ITS4 primers [21]. Amplifications were performed on a reaction volume of 25 μL containing 22 μL of ABGene 1.1 × Master Mix, 2.5 mM MgCl2 (Thermo Scientific, Waltham, MA, USA), 0.5 μL of 0.4% bovine serum albumin (BSA), 0.5 μL of dimethyl sulfoxide (DMSO), 0.5 μL of each primer (10 pmol/μL), and 1 μL of template DNA on a 9700 GeneAmpl thermocycler (Applied Biosystems, Waltham, MA, USA). The PCR program for ITS was: 2 min at 95 °C, followed by 30 cycles of 95 °C for 1 min, 53 °C for 1 min, 72 °C for 2 min and a final extension at 72 °C for 5 min.

Sequencer 4.1 (Gene Codes Corp., Ann Arbor, MI, USA) was used to assemble complementary strands and verify software base-calling. Sequence alignment was performed using MUSCLE [22] conducted in MEGA X v.10.2.6 [23] with minor manual adjustments to get the final aligned matrix. Twenty-eight ITS sequence data belonging to 16 species of *Frankenia*, covering most of the African and Mediterranean groups accepted in the genus, were used in the analyses, they all being obtained specifically for this study (Table 2). GenBank accessions of *Tamarix gallica* L. (code MH626294), *Myricaria germanica* L. (code KJ808607) and *Reaumuria alternifolia* (Labill.) Britten (code KJ729627) were used as outgroups. It is worth mentioning that the name *R. alternifolia* is used here instead of its superfluous synonym *R. hypericoides* Willd. that is applied in GenBank to the sequence used.

Phylogenetic analyses of the ITS region were obtained using Maximum Parsimony (MP), Maximum Likelihood (ML), and Neighbour Joining (NJ) methods. MP analysis was conducted in both PAUP (using Heuristic search options with the tree searching strategy based on Nearest Neighbour Interchange, NNI) and MEGA (using Heuristic search options with the tree searching strategy based on Subtree-Pruning-Regrafting –SPR– with search level 1 [24]) for result comparison with 10,000 replicates. ML [25] and NJ [26] analyses were also performed in MEGA as well as the selection of the best model of DNA substitutions for each method using the Akaike Information Criterion (AIC; [27]); models with the lowest BIC (Bayesian Information Criterion) scores were considered to best describe the substitution pattern for the ML and NJ analyses. Phylogenetic reconstructions for ML and evolutionary distances for NJ were estimated using the K2 model (2-parameter method of Kimura [28]) and considering all sites; the rate variation model allowed for some sites to be evolutionarily invariable (+*I*, 27.37% sites). For comparison purposes, remotion of all ambiguous positions for each sequence pair (Pairwise Deletion option) was also performed, and no significant differences (only affecting BP values in a few branches) were observed in the obtained phylogenies. For all those methods, support was assessed by the bootstrap [29] with 10,000 replicates but holding only 10 trees per replicate. Clades showing bootstrap percentage (BP) values of 50–74% were considered weakly supported, 75–89% moderately supported, and 90–100% strongly supported.

**Table 2 plants-12-02630-t002:** List of outgroups and *Frankenia* accessions used for the ITS phylogenetic analyses.

Taxon	Provenance (Herbarium Voucher)	Source	GenBank Code
*Tamarix gallica* L.	France: Saintes Maries de la Mer (ABH57865)	Villar et al. [30]	MH626294
*Myricaria germanica* L.	Kazakhstan: Zajsanskaya depression (LE)	Zhang et al. [31]	KJ808607
*Reaumuria alternifolia* (Labill.) Britten ^1^	Azerbaijan: Caucasus (MW)	Zhang et al. [32]	KJ729627
*Frankenia anneliseae* M.B.Crespo & al.	South Africa: Klipfontein (ABH76891)	This paper	OR183455
	South Africa: Skoverfontein (ABH83196)	This paper	OR183456
*F. boissieri* Reut. ex Boiss.	Spain: Huelva, Ayamonte, Is. Canela (ABH83543)	This paper	OR183457
*F. capitata* Webb & Berthel.	Spain: Gran Canaria, Isleta (ABH83612)	This paper	OR183458
*F. composita* Pau & Font Quer	Morocco: Al Hoceïma, Cala Iris (ABH81590)	This paper	OR183459
*F. corymbosa* Desf.	Morocco: Al-Hoceïma (ABH54256)	This paper	OR183460
	Morocco: Nador, Punta Charrana (ABH54294)	This paper	OR183461
	Spain: Alicante, Santa Pola (ABH79956)	This paper	OR183462
	Spain: Murcia, Cabo Cope (ABH83531)	This paper	OR183463
*F. ericifolia* C.Sm. ex DC., nom. cons. prop.	Spain: Tenerife, Punta de Teno (ABH79975)	This paper	OR183464
	Spain: Tenerife, Güímar (ABH83613)	This paper	OR183465
*F. fruticosa* J.C.Manning & Helme.	South Africa: Moedverloren (ABH76898)	This paper	OR183466
*F. hirsuta* L.	Türkiye: Tuz Gölii, salty lagoon (ABH45933)	This paper	OR183467
*F. ifniensis* Caball.	Morocco: Sidi Ifni to Oued Noun (MA758515)	This paper	OR183468
*F. laevis* L.	France: Aude, Étang de La Palme (ABH70584)	This paper	OR183469
	Italy: Sardinia, Cagliari (ABH70072)	This paper	OR183470
*F. nummularia* M.B.Crespo & al.	South Africa: Kookfontein River (ABH83290)	This paper	OR183471
	South Africa: Tankwa Karoo (ABH83295)	This paper	OR183472
*F. pseudoericifolia* Rivas Mart. & al.	Portugal: Cape Verde, São Antão (MA0906845)	This paper	OR183473
*F. pulverulenta* L.	South Africa: Redelinghuys (ABH77205)	This paper	OR183474
	Spain: Teruel, Alcañiz (ABH73564)	This paper	OR183475
	Spain: Alicante, Cabo de las Huertas (ABH41853)	This paper	OR183476
	Spain: Tenerife, Puerto de la Cruz (ABH79974)	This paper	OR183477
*F. repens* (P.J.Bergius) Fourc.	South Africa: S of Groenrivier (ABH76868)	This paper	OR183478
	South Africa: S of Hondeklipbaai (ABH76862)	This paper	OR183479
*F. thymifolia* Desf.	Spain: Zaragoza: Bujaraloz (ABH75454)	This paper	OR183480
	Algeria: Bougtob, Chott Cherguí (ABH59344)	This paper	OR183481
*F. velutina* Brouss. ex DC.	Morocco: Essaouira (ABH79929)	This paper	OR183482

^1^ Filed in GenBank as the synonym name *R. hypericoides* Willd. (nom. illeg.).

Furthermore, Bayesian inference (BI) analyses were conducted with MrBayes 3.2 [33], in which the Markov Chain Monte Carlo (MCMC) algorithm was run for 10 million generations and sampled every 1000 generations. Two runs were performed. The general time reversible (GTR) + proportion of invariant sites (*I*) + gamma distribution (*G*) model was used in the analyses (set nst = 6 rates = invgamma) according to the results obtained with jModelTest 2.1.10 [34] under AIC. The first 25% generations (burninfrac = 0.25) were excluded, and the remaining trees were used to compile a posterior probability (PP) distribution using a 50% majority rule consensus.

## 3. Results

### 3.1. A Brief Story and Typification of Frankenia nodiflora

Lamarck [13] included four species in his treatment of *Frankenia* for the *Encyclopédie Méthodique*. Among them, he described *Frankenia nodiflora* (numbered 3) as new after *F. hirsuta* L. (numbered 2) and before *F. pulverulenta* L. (numbered 4). No illustrations were cited in the protologue, though the new species was described from dried material as follows: “3. FRANQUENNE nodiflore, *Frankenia nodiflora*, *Frankenia caulibus simplicibus filiformibus subglabris*, *fasciculis florum lateralibus axillaribus* & *oppositis.* N[obis]. Ses tiges sont longues de six à huit pouces, simples ou presque simples, filiformes, feuillées, & presque glabres. Leurs feuilles sont opposées, pétiolées, ovales, glabres, à bords réfléchis en dessous, & longues de deux lignes & demie. Dans leurs aisselles, on observe sur toute la longueur des tiges, des rameaux non développés, opposes, plus courts que les entre-noeuds, feuillés & fleuris, & qui sont paroître les tiges entrecoupées dans leur longueur par des touffes ou paquets de feuilles & de fleurs biens sépares les uns des autres. Les fleurs ont leur calice oblong, légèrement anguleux, glabre, & naissent comme en faisceau entre les feuilles, aux noeuds des tiges. Cette plante croît naturellement au Cap de Bonne-Espérance, & a l’aspect d’une espèce de Salicaire. (*v. s.*)”. Furthermore, in the comment for *F. hirsuta*, he clearly indicated the origin and collector of the material used for the description of *F. nodiflora*: “La plante β [of the intended “*F. hirsuta*”, probably corresponding to *F. repens*] a ses tiges presqu’entièrement glabres, ainsi que ses calices; elle croît au Cap de Bonne-Espérance, & nous a été communiquée avec la suivante [*F. nodiflora*] par M. Sonnerat”. In fact, the French naturalist and explorer Pierre Sonnerat (1748–1814) most probably gathered that material in the surroundings of Cape Town during his travels to the East Indies and China in 1774–1781. The results of those expeditions were published later in a two-volume work [35], which includes observations on the Cape area in the second volume that covers his visits to Cape Town, Madagascar, the Maldives, Mauritius, Ceylon (Sri Lanka), Indonesia, Burma, China, and the Philippines.

*Frankenia nodiflora* was later depicted in Plate 262 of Lamarck’s *Tableau encyclopédique et méthodique* [36] together with *F. pulverulenta* (Figure 1), though the details are poor. He also added a brief diagnosis in the French and Latin languages, with slight changes with regard to the protologue in the branching pattern of the stem: “4. FRANQUENNE nodiflore. Dict. n° 3. F. tiges simples, filiformes, presque glabres; fascicules des fleurs latéraux, axillaires et opposés. [FRANKENIA nodiflora. F. caulibus simplicibus, filiformibus, subglabris; fasciculis florum lateralibus, axillaribus et oppositis.]”.

Details in the protologue of *F. nodiflora* on the length and branching pattern of the type material (“Ses tiges sont longues de six à huit pouces, simples ou presque simples…”) point to the existence of various vouchers on which the description was prepared. In the herbarium P, we have traced two specimens matching the protologue, which are relevant for typification. First, the voucher P00287094! (Figure 1a) in Lamarck’s herbarium is labelled “*Frankenia nodiflora* Lam./dict./e cap[ut]. b[onae]. Sp[ei].” by Lamarck himself and includes a single unbranched fragment of ca. 15 cm long (ca. 6 inches). Secondly, the voucher P05038792!, which is marked as part of Maire’s herbarium, bears a label reading “*Frankenia nodiflora* Lam./(ego.)/Cap de B[onne]. Espér[ance]” in a calligraphy not incompatible with Lamarck’s handwriting and includes two unequal fragments: one of ca. 19 cm long (ca. 7.6 inches) with a short lateral branch in the upper part, similar to that illustrated in Lamarck [36], and the other of ca. 4 cm long (ca. 1.6 inches). Both specimens bear fragments very similar to each other, more likely coming from a single collection, and therefore, they might be regarded as duplicates belonging to the original material of that name. Because the specimen P00287094 is in Lamarck’s collection, it is designated below as lectotype for *F. nodiflora*; the specimen P05038792, despite some doubts about the handwriting on its label being Lamarck’s (C. Aupic pers. comm.) and why it was placed among Maire’s material, is here regarded as isolectotype. Lamarck’s concept of *F. pulverulenta* can be inferred not only from the cited plate 262 [36] (Figure 1b), but also from some sheets in his personal herbarium at P (P00287095! and P00287096!; available online: https://science.mnhn.fr/institution/mnhn/collection/p/item/search/form?lang=fr_FR; accessed on 3 April 2023), which match the Linnaean type of that species.

Among the diagnostic characters in the protologue summarised by Lamarck [13,36], the long internodes, the condensed glomerular inflorescences, the ovate petiolate, subacute and entirely glabrous leaves, and the angulose and glabrous calyces are differential for *F. nodiflora*. The revision of herbarium material at K and P revealed the existence of plants occurring in the surroundings of Cape Town, which perfectly match Lamarck’s type material. They are perennial shrublets producing suffruticose procumbent, non-rooting stems (or rooting only at the base) with long internodes; leaves entirely glabrous, mostly concolorous (bright green on both sides), broadly elliptic to oblong, flattened and only folded downwards on margins (at least on the upper third), mostly falcate upwards, subacute to minutely mucronate at apex, fleshy, with conspicuous petiole 0.6–1.2 mm long, sometimes glabrous; bracteoles broad and flat, about half to two thirds the length of the calyx; flowers mostly disposed in crowded dichasial glomerules on lateral short branches; calyxes often glabrous, with teeth cucullate bearing a notable subapical mucro ca. 0.5 mm long (diagnostic character not present in other African taxa of the genus); and seeds ca. 1 mm long, covered with unequal medium-sized papillae 12–33 μm long, globose to conical–obtuse, more densely disposed on the distal part, among other characters.

*Frankenia nodiflora* has been treated in quite different ways. Candolle [37] accepted it as a distinct species, but Harvey [38], who did not see Lamarck’s specimens, suggested its probable inclusion in *F. pulverulenta* perhaps as a variety; in the same work, he surprisingly also regarded the South African “*F. pulverulenta* & *F. nodiflora*, of Drège’s Coll.”, which included other gatherings of the true *F. nodiflora* (conserved in different European herbaria; see below) to belong to *F. pulverulenta*. Similarly, *F. nodiflora* was often considered to be merely a synonym of *F. pulverulenta* [1,4,39], and subsequently, recent works did not accept Lamarck’s species in the Southern African floras [10,11,40,41], probably assuming implicit synonymisation with *F. pulverulenta*. That synonymic treatment is currently accepted in POWO [12] under the name *F. pulverulenta* subsp. *pulverulenta* (https://powo.science.kew.org/taxon/urn:lsid:ipni.org:names:77229962-1#synonyms; accessed on 3 April 2023).

Nevertheless, in our view, the Cape plants matching the Lamarckian concept of *F. nodiflora* are morphologically distinct from the Linnaean *F. pulverulenta*, showing a unique combination of characters unknown in the other Southern African species of *Frankenia*, this supporting acceptance at specific rank. Other similar populations growing in the northern and western inland areas of the Nama-Karoo and Succulent-Karoo biomes that show some resemblances to *F. nodiflora* and were often misidentified as *F. pulverulenta*, *F. capitata*, or *F. repens* (incl. *F. krebsii*), are here re-evaluated in the light of new morphological and molecular data.

### 3.2. Phylogenetic Relationships

The aligned ITS dataset was 718 bp, 192 of which (26.74%) were potentially parsimony informative. The phylogenetic relationships of taxa are shown in Figure 2 as recovered in our BI and ML trees. PP values from the BI analysis are shown above branches in the BI tree, whereas the percentage of trees in which the associated taxa clustered together (after 10,000 bootstrap replicates) are shown above branches in the ML tree. Analyses of this dataset, using MP and NJ methods, yielded trees with similar topologies to BI and ML trees, respectively (see Appendix A). In the MP analysis, nine most parsimonious trees were obtained with a tree length (TL) of 516 steps, a consistency index (CI) of 0.769, and a retention index (RI) of 0.863.

Species of *Frankenia* are arranged in three strongly supported main groups, though not fully resolved. First, clade A (1.00 PP, 96 BP) contains the South African members related to *F. nodiflora* nested in a strongly supported clade (1.00 PP, 98 BP), together with *F. repens*. This three-species group (hereafter the “*F. repens* group”) includes three well to strongly supported lineages that correspond to the newly described *F. nummularia* (1.00 PP, 90 BP) and *F. anneliseae* (1.00 PP, 99 BP) plus *F. repens* (1.00 PP, 99 BP). This clade is sister (0.99 PP, 90 BP) to a group including the E Iberian–W Algerian endemic *F. thymifolia* (1.00 PP, 96 BP) plus the SW Iberian–NW Moroccan endemic *F. boissieri* Reut. ex Boiss. Second, clade B (1.00 PP, 99 BP) includes taxa with a broad Mediterranean, Macaronesian, Atlantic, or Subcosmopolitan distributions. It is weakly supported as sister to clade C in the BI tree (0.77 PP), but its position is not resolved in the ML tree. Two subclades are recovered in our analyses. On the one hand, a moderately supported subclade (0.74 PP, 79 BP) groups the Macaronesian and Western Saharan endemics of the genus. The Canarian *F. ericifolia* C.Sm. ex DC. (1.00 PP, 99 BP) is weakly sister in the ML tree (59 BP) to a strongly supported group (1.00 PP, 100 BP), also including the Canarian *F. capitata* plus the W Saharan *F. ifniensis* Caball., a relationship not recovered in the BI tree. The Cape Verdean endemic *F. pseudoericifolia* Rivas Mart. & al. is sister to the members of this subclade only in the ML tree (0.79 BP), but its position is unresolved in the BI tree. On the other hand, a strongly supported subclade (1.00 PP, 99 BP) includes samples of the annual *F. pulverulenta* from South Africa, the Canary Islands, and the Iberian Peninsula that form in the ML analysis a moderately supported clade (89 BP), which in the ML tree is moderately supported (78 BP) as sister to a clade that includes western Mediterranean–Atlantic (*F. laevis* L. and *F. composita* Pau and Font Quer) and northwestern Atlantic Moroccan (*F. velutina* Brouss. ex DC.) members of the genus. Finally, clade C (0.99 PP, 99 BP) is formed by accessions of *F. corymbosa* Desf. from SE Iberian Peninsula and N Morocco, though the internal relationships of this clade are very weak and are not connected with a geographical provenance of samples. The eastern Mediterranean species *F. hirsuta* L. (s.l.) and the outstanding South African narrow endemic, *F. fruticosa*, are successive sisters to the rest of the studied taxa (clade A + B + C), relationships that are in need of further investigation.

### 3.3. Number of Ovules and Seed Morphology

Features of ovules and seeds are an important source of diagnostic taxonomic data for the studied species of *Frankenia*, mostly regarding the number of ovules per placenta, seeds per capsule, seed size, and ornamentation of the testa surface (Figure 3). Capsules of the studied South African species of *Frankenia* show three carpels and three placentas with a number of ovules ranging from 2 (in *F. fruticosa*), 5–10 (in the “*F. repens* group”), or 13–18 (in *F. pulverulenta*) in each one. In general terms, seeds are typically ellipsoid to ovoid–ellipsoid, conspicuously sulcate on one side (raphe), and pale brown in colour but darker at the funicular part, and they develop rapidly even before the flower has completely withered. Two trends have been observed concerning the number and size of mature seeds: (i) numerous (up to 45 per capsule) and smaller (ca. 0.5–0.7 mm long) in the subcosmopolitan *F. pulverulenta* and (ii) less numerous (up to 22 per capsule) and larger (ca. 0.7–1.1 mm long) in the “*Frankenia repens* group”. However, differences exist among the three South African members of that group. On the one hand, both *F. nodiflora* and *F. nummularia* generated 6–10 ovules per placenta, but only 9–12 seeds were counted per capsule in the former, whereas 12–22 seeds per capsule were observed in the latter. On the other hand, *F. anneliseae* produced a lower number of ovules (5–6) per placenta and also yielded a much lower number of seeds per capsule (1–8).

The testa of seeds is thin, not mucilaginous, with a surface weakly and irregularly ornamented with a subrectangular-reticulate pattern, finely striated, and covered with papillae that can vary in morphology, size, and distribution. First, *F. pulverulenta* shows small-sized (4–17 μm long) papillae, homogeneous (conical-obtuse) in shape, and very sparsely covering the testa surface (Figure 3a,b). Secondly, *F. nodiflora* and *F. anneliseae* exhibit medium-sized papillae (respectively, 10–43 μm and 12–33 μm long) that are heterogeneous (globose to conical-obtuse) in shape but more densely disposed on the distal part (Figure 3c,d,g,h). Similarly, *F. nummularia* produces almost smooth seeds, only sparsely covered with small papillae (3.5–9 μm long) on the funicular side, mostly homogeneous (conical–obtuse) (Figure 3e,f). Finally, *F. repens* differs markedly from the remaining South African members of the “*F. repens* group” since its seeds display large-sized papillae (55–110 μm long), are almost homogeneous (cylindrical-conical obtuse) in shape, and are densely covering the testa surface (Figure 3i,j). Those patterns were constant in the observed samples, and no intermediates were found among the described types.

### 3.4. Taxonomic Treatment of Frankenia nodiflora and Description of New Related Species

#### 3.4.1. ***Frankenia nodiflora*** Lam., Encycl. 2(2): 543. 1788 ≡ *Franca nodiflora* (Lam.) Vis. in Mem. Reale Ist. Veneto Sci. 16: 158. 1871.

*Type*: [South Africa. Western Cape]. E Cap[ut]. B[onae]. Sp[ei]. [*Sonnerat* s.n.] (**lecto. designated here**: P00287094!, Figure 1a; isolecto.: P05038792! (Cap de B[onne]. Espér[ance]. [*Sonnerat* s.n.]).

-*F. pulverulenta* auct. pl. atque *F. krebsii* auct. pl.

*Description*: *Shrublet* densely branched, tap-rooted, woody at the base with grey bark, and mostly glabrous. The *stems* are rooting only at base, diffuse, usually creeping, suffruticose, up to 40 cm long, and often with divaricate branches 1–3 cm long; the *young branchlets* show internodes up to 35 mm long, yellowish to reddish, and glabrous or sometimes shortly and loosely puberulous. The *leaves* are opposite-decussate, patent to erect-patent, bright-green or sometimes reddish, and glabrous on both sides; the *petiole* is 0.6–1.2 × 0.2–0.3 mm, flattened, and tapering distally; the *sheath* extends along margins of the petiole to the blade, is laxly ciliate (occasionally subglabrous), with 2–5 pairs of lateral cilia 0.2–0.8 mm long, unequal, whitish, flattened, and obtuse or acute at the apex; the *leaf blade* is 2.5–3.5 × 1–2 mm, broadly ovate-elliptic to oblong, mostly falcate upwards, subacute to minutely mucronate at the apex, fleshy, concolorous, sometimes slightly paler abaxially with minute glands bearing salt depositions, often cochleariform (convex adaxially) to flattened, longitudinally folded downwards, with margins strongly to loosely (and gradually) revolute at least on the upper third, and somewhat thickened; the *midrib* is narrow, linear, tapering slightly towards apex, continuous with petiole below, somewhat raised abaxially, and extending all along the blade length; the *young leaves* are on short shoots, fasciculate, and similar but smaller than those on long shoots and sometimes narrower. The *flowers* are pentamerous, perfect, borne commonly in dichasial groups, usually condensed at stem nodes, glomerular or with short axillary branches up to 20 mm long (usually shorter); the central flower is sessile, and the lateral ones are on pedicels up to 1 mm long. The *floral bracts* are two in number, 2–3 mm long, leaf-like, subpatent to erect-patent, connate at the base, enveloping the basal part of the calyx for 0.5–0.7 mm. The *bracteoles* are two in number, 1–2 mm long, bract-like but smaller, about half to two thirds the length of the calyx, adnate to the calyx base, and alternating with bracts. The *calyx* is 4–4.5 × 1–1.5 mm, tubular at the anthesis, untwisted, straight, indurate, with five prominent thickened ribs, and entirely glabrous; the *teeth* are five in number, 0.9–1.2 mm long, triangular, narrowly membranous and shortly papillate on the margins, often reddish, slightly recurved after anthesis, cucullate with a notable subapical acumen ca. 0.7 mm long, obtuse, and slightly divergent. The *petals* are five in number, 5–7 × 1–2 mm, obovate-cuneate, pinkish-mauve but whitish below, and overlapping only in the basal part; the *claw* is 2–3 × 0.5–0.7 mm, narrowly cuneate, imperceptibly tapering to blade, yellowish, and hidden into the calyx tube; the *ligule* is 1.5–2.5 × 0.3–0.4 mm, narrowly oblanceolate, longitudinally adnate to the claw, the free apex ca. 0.5 mm long, oblong-obtuse, and entire; the *blade* is 2.3–3 × 1.5–2 mm, broadly obovate with apex rounded to truncate, slightly emarginate, and irregularly sinuate (not erose-denticulate). The *stamens* are six in number, in two unequal whorls, usually long exserted, and overtopping 1.5–2 mm the calyx teeth at anthesis; the *filaments* are 3–6 mm long, expanded ca. 0.5 mm wide in the lower half but gradually tapering and filiform in the distal half, and whitish; the *anthers* are 0.5–0.7 mm long, oblong-ellipsoid, versatile, and yellowish. The *ovary* is ellipsoid, subtrigonous, and with three carpels; the *placentae* are three in number, parietal-basal, extending up to the lower half to two thirds of carpel wall length, and have ventral traces moderately to highly branched; the *ovules* are 4–6 per placenta and are attached along most of the placenta by erect funiculi 0.1–0.2 mm long. The *style* is 3–4.5 mm long, terete, somewhat sigmoid at the base, exserted and elongated up to 8 mm after pollination, and whitish; the *style branches* are three in number, filiform, 0.5–0.7 mm long, and whitish; the *stigmas* are slightly clavate. The *capsule* is 2.5–3.5 × 0.6–2 mm, ovoid-ellipsoid, hidden in the calyx tube, dark reddish-brown, and early dehiscent. The *seeds* are 9–12 per capsule, 0.7–1.0 × 0.3–0.5 mm, sulcate on one side, ellipsoid, pale brown, darker at the funicular part, and developing rapidly even before the flower has completely withered; the *testa* is thin, not mucilaginous, with a surface weakly and irregularly ornamented with a subrectangular-reticulate pattern, finely striate, covered with medium-sized papillae 12–33 μm long, heterogeneous, globose to conical-obtuse, are more densely disposed on the distal part.

*Etymology*: The specific epithet (*nodiflorus, –a, –um* = with flowers at nodes) refers to the disposition of flowers and inflorescences, mostly crowded at stem nodes.

*Phenology*: The species flowers in late October–early January (occasionally in July–August) and fruits occur in November–February (occasionally in August–September).

*Habitat and distribution*: *Frankenia nodiflora* occurs on saline, azonal soils of salt pans and saline riverbeds in the coastal lowlands at elevations of 0–150 m above sea level. The known distribution of the species extends across the Cape Flats into neighbouring areas northeast from Cape Town in the Western Cape Province (Figure 4), a territory broadly included in the Fynbos (F) biome (mostly the F07 Bioregion) *sensu* Mucina and Rutherford [14], where it specifically inhabits the so-called “Cape Inland Salt Pans” (code AZi 9). In the coastal lowlands of the Fynbos biome, the climate is mild and oceanic (ameliorated by the ocean influences), with average temperatures ranging about 7 °C in winter to 30 °C in summer (average annual temperature ca. 17 °C) and frosts being rare and occasional. The average annual precipitation amounts to ca. 500–540 mm, though rather differently distributed, the rainfall occurring mostly during winter (May to August) with a peak in July [14].

*Notes*: In the past, *F. nodiflora* was surely more abundant than today before the severe changes that extensive agriculture and urbanization have brought to the Cape Flats landscape. The three known populations of this species are restricted to nature reserves at a few sites between Durbanville (Figure 5) and Paarl where meadows and patches of halophytic vegetation are being conserved. The most important threats to this species are related to the negative effect of alien invasive species, such as *Cynodon dactylon* L. and *Cenchrus clandestinus* (Hochst. ex Chiov.) Morrone (*Pennisetum clandestinum* Hochst. ex Chiov.) (P. Winter pers. comm.), on natural habitats. In this context, urgent field prospections are required to locate new populations of this rare endemism that have probably gone unnoticed, and the active management of natural sites is also required to prevent the decline and extinction of the few wild populations. Therefore, new information is needed for accurate conservation labelling of the species, including counts of the number of populations and individuals as well as their evolution over time. Meanwhile, the conservation status of *F. nodiflora* is suggested here as Data Deficient (DD), although it probably might be assessed as at least Endangered (EN), according to IUCN [42] since the extant populations are found in an estimated Extent of Occurrence (EO) of ca. 180 km^2^ and an Area of Occupancy (AO) smaller than 5 km^2^, with presumably severe population fragmentation and reduction of the habitat quality.

*Other studied materials*: South Africa. **Western Cape Province**: 3318 (Cape Town): Cape of Good Hope, Table Mountain (−CD), December 1832, *J. Mac Gillivray 584* (K!); 3318 (Cape Town): Cape Town, Green Point (−CD), November 1846, *A. Prior* s.n. (K!); 3318 (Cape Town): Cape Town, Mowbray (−CD), shores of vlei, August 1912, *W.C. Worsdell* s.n. (K!); 3318 (Cape Town): Cape Peninsula, Raapenberg Vley (−CD), 26 November 1896, *A.H. Wooley 2110* (BOL!, K!, excl. fragment in the central part); 3318 (Cape Town): Cape Town, about Salt River near the Windmills (−CD), 14 November 1811, [Pl. Africae Australis Extratropicae,] *Burchell 513* (G-DC G00211140!, K!, P05038790!). 3318 (Cape Town): Cape Town, Uitkamp Wetlands Nature Reserve (−DC), 33°48′59.7″ S 18°38′26.0″ E, 137 m elev., 7 November 2014, *H. Stummer* s.n. (ABH83529!).

*Unidentified sites*: South Africa. Habitat ad C[aput] B[onae] Spei (B-W06993). E Cap[ut]. B[onae]. Sp[ei], [*Sonnerat* s.n.] (P00287094!); Cap de B[onne]. Espér[ance], [*Sonnerat* s.n.] (P05038792!). Cape, *Dr. Pappe* s.n. (K000232047!). Cap de Bonne-Espérance, 1842 (MPU693060!). Cap de Bonne-Espérance (P05038793!). C[aput] B[onae] S[pei], *Mrs. Gilavray* s.n. (P05038816!). Cap de Bone Espérance, ex herb. *Pet. Thouare* s.n. (P05038804!). [Cape] Pl. Capenses, *Ecklon* (P05038764!). No. 1380, 1772, *Oldenburg* (BM!).

*Digital iNaturalist images*: South Africa. **Western Cape Province**. 3318 (Cape Town): Cape Town, Durbanville, Belleville, Uitkamp Wetlands Nature Reserve (−DC), 33°48′59.7″ S 18°38′26.0″ E, 137 m elev., 7 November 2014, *M.Goets* (v.v.): https://www.inaturalist.org/observations/11060535, accessed on 3 April 2023; 3318 (Cape Town): Cape Town, Malmesbury Farms, Teleport Rd (−DA), 33°41′18.0″ S 18°42′25.0″ E, 88 m elev., 26 April 2019, *I. Ebrahim* (v.v.): https://www.inaturalist.org/observations/24159910, accessed on 3 April 2023; 3318 (Cape Town): Paarl, Cape Winelands District, Windmeul Farm, Langerug Private Nature Reserve (−BD), 33°39’37.14″ S 18°54’1.62″ E, 146 m elev., 10 November 2017, *J. Wicht* (v.v.): https://www.inaturalist.org/observations/148456049, accessed on 3 April 2023.

#### 3.4.2. ***Frankenia nummularia*** M.B.Crespo, M.Á.Alonso, Mart.-Azorín, J.L.Villar & Mucina, **sp. nov.**

*Type*: South Africa. Western Cape: Karoo, Beukesfontein, sandy river bed, 1420 ft elev., 1 December 1908, *H.H.W. Pearson 5005* (holo.: BOL!; iso.: BM!, K!, P05038725!). Figure 6a.

-*F. pulverulenta* auct. pl. atque *F. repens* auct. pl.


*Diagnosis: Planta speciosa a F. pulverulenta et F. nodiflora foliis latis, subplanis (vel ad margines parve revolutis), longe petiolatis, et caulibus prostratis vel ascendentibus accedenti, sed ab eas distinctissima et bene distinguenda. A priore insuper differt caulibus suffruticosis, perennibus (non herbaceis annuis); foliis glaberrimis (non subtus pilosis); bracteolis calycem multo brevioribus (non aequantibus); et seminibus minus numerosis (ad 22, non 45) minoribusque (0.7–1.1 mm long., non 0.5–0.7 mm long.). A posteriore insuper discrepat foliis discoloribus, subtus valde pallidioribus (non subconcoloribus); calycibus majoribus 4.5–6 mm long., dentibus acutis vel mucrone inconspicuo ad 0.2 mm long. (non calycibus 4–4.5 mm long., dentibus mucrone magno ad 0.7 mm long.); et praesertim seminum testa papillis brevioribus, 3.5–9 μm long., subhomogeneis, omnibus conico-obtusis (non papillis longioribus 12–33 μm long., heterogeneis, aliis globosis aliis conico-obtusis).*


*Description*: *Shrublet* densely branched, tap-rooted, woody at the base with grey bark, and glabrous to long hairy. The *stems* are rooting only at the base, procumbent to ascending, suffruticose, up to 40 cm long, and usually with erect, often divaricate branches 5–35 cm long; the *young branchlets* show internodes up to 30 mm long, yellowish to reddish, glabrous to densely hairy, covered mostly on one side with heterogeneous indumentum of both curled minute trichomes (ca. 0.2 mm long) and flexuous long pluricellular complanate trichomes (0.7–1 mm), and denser below the nodes. The *leaves* are opposite-decussate, patent to erect-patent, greyish-green to bright green, somewhat glaucescent, glabrous on both sides, and sometimes with salt depositions; the *petiole* is 0.7–1.5 × 0.3–0.5 mm, flattened, and tapering distally; the *sheath* extends along margins of petiole almost to the blade, densely ciliate (occasionally almost glabrous), with 2–5 pairs of lateral cilia 0.3–0.8 mm long, unequal, and whitish; the *leaf blade* is 2–7 × 1.5–6 mm, mostly suborbicular or broadly elliptic, rounded to slightly pointed at the apex, somewhat fleshy, often discolorous, abaxially paler with minute glands usually bearing salt depositions, often longitudinally folded, with margins loosely and gradually revolute, and somewhat thickened; the *midrib* is narrow, linear, tapering slightly towards apex, continuous with the petiole below, somewhat raised abaxially, and extending about half the blade length; the *young leaves* are on short shoots, fasciculate, similar but smaller than those on long shoots, and sometimes narrower and incurved on margins at the upper part. The *flowers* are pentamerous, perfect, borne commonly in loose dichasial groups, usually widely branched, and with erect-patent branchlets up to 20 mm long; the *lowermost flowers* are solitary and sessile, and the uppermost dichasia are often denser and subcorymbose (the central flower always sessile, the lateral ones on pedicels 1–2 mm long). The *floral bracts* are two in number, 2–7 mm long, leaf-like, subpatent to erect-patent, connate at the base, and enveloping basal part of calyx for 0.5–0.7 mm. The bracteoles are two in number, 1–2 mm long, bract-like but smaller, up to half the length of the calyx, adnate to the calyx base, and alternating with bracts. The *calyx* is 4.5–6 × 1–1.5 mm, fusiform-tubular to gradually fusiform after anthesis, untwisted, straight to slightly curved, indurate, with five prominent thickened ribs, entirely glabrous or sparsely puberulous to hirtellous at base and on ribs, with hairs up to 0.5 mm long, whitish, and occasionally with scattered whitish depositions on the upper part; the *teeth* are five in number, 0.7–0.9 mm long, triangular, acute to minutely mucronate (mucro up to 0.2 mm long), minutely papillate on margins, often yellowish, and erect to slightly incurved after anthesis. The *petals* are five in number, 6–9 × 1–2 mm, long obovate cuneate, pinkish-mauve, and overlapping in most of their length; the *claw* is 4–5 × 0.8–1 mm, narrowly cuneate, imperceptibly tapering to blade, whitish, and hidden in the calyx tube; the *ligule* is 2–3 × 0.5–1 mm, narrowly oblanceolate, longitudinally adnate to claw, the free apex ca. 1 mm long, triangular-acute, and entire; the *blade* is 2.3–4 × 1.5–2 mm, broadly obovate to suborbicular with apex rounded to truncate, and irregularly erose-denticulate. The *stamens* are six in number, in two unequal whorls, exserted, and overtopping 1.5–2 mm the calyx teeth at anthesis; the filaments are 6–8 mm long, expanded ca. 0.5 mm wide in the lower half but gradually tapering and filiform in the distal half, and pinkish to bluish-pink; the *anthers* are 0.8–1.2 mm long, ellipsoid, versatile, and yellowish to reddish. The *ovary* is ellipsoid, subtrigonous, and with three carpels; the *placentae* are three in number, parietal-basal, extending up to the lower half to two thirds of carpel wall length, and with ventral traces moderately to highly branched; the *ovules* are 6–10 per placenta and attached along most of placenta by erect funiculi 0.3–0.4 mm long. The *style* is 6–9 mm long, terete, somewhat sigmoid at the base, exserted and elongating up to 11 mm after pollination, and whitish; the *style branches* are three in number, filiform, 0.5–0.7 mm long, and pinkish-white to reddish; the *stigmas* are slightly clavate. The *capsule* is 2.5–3.5 × 1–2 mm, ovoid-ellipsoid, hidden in the calyx tube, dark reddish-brown, and early dehiscent. The *seeds* are 12–22 per capsule, 0.7–0.9 × 0.3–0.5 mm, ellipsoid to ovoid-ellipsoid, sulcate on one side, pale brown, darker at the funicular part, and developing rapidly even before flower has completely withered; the *testa* is thin, not mucilaginous, with surface weakly and irregularly ornamented with a subrectangular-reticulate pattern, finely striate, almost smooth, only sparsely covered on the funicular side with small papillae 3.5–9 μm long, subhomogeneous, and conical-obtuse.

*Etymology*: The specific epithet (*nummularius*, –*a*, –*um* = coin-bearing) refers to the shape of leaves and bracts, which are usually suborbicular and flattened, resembling coins.

*Phenology*: The species flowers in late October–early January (occasionally in July–August) and fruits occur in November–February (occasionally in August–September).

*Habitat and distribution*: *Frankenia nummularia* grows in saline, well-drained sandy, azonal dry soils of salt pans, beds of intermittently flowing (occasionally during regional precipitation-rich events) semi-desert rivers and streams in inland regions, and at elevations of 250–1380 m above sea level (Figure 6b). The known distribution of the species extends through most of the SW part of the Karoo Region in western South Africa, ranging from Vanrhynsdorp and Riversdale District in the Western Cape to Calvinia and Victoria West District in the Northern Cape Province (Figure 4), a territory mostly included in the Nama-Karoo (NK) biomes and reaching the southern Succulent-Karoo biomes (mostly the SKk, SKt and SKv Bioregions) *sensu* Mucina and Rutherford [14], where it occurs in the so-called “Bushmanland vloere” (code AZi 5). In those regions, the climate is semiarid to arid, continental (not or scarcely ameliorated by the ocean influences), and with average temperatures ranging from −5 °C in winter to 43 °C in summer and frosts being usual in the higher areas. The average annual precipitation varies between 100 mm and 500 mm, though rather differently distributed, the rainfall occurring mostly during late summer (December to April) with a peak in March [14].

*Notes*: Wild populations of *F. nummularia* include numerous individuals covering a large territory in South Africa, and no special threats are known so far to occur that might lead to any inferred decline in either the number of populations or the number of individuals. Therefore, its conservation status is suggested here as Least Concern (LC) according to IUCN [42].

*Other studied materials*: South Africa. **Northern Cape Province**: 3021 (Vanwyksvlei): Rietspoort (−CB), (I, d1), 3000–4000 ft elev., 30 November 1826, *J.F. Drège 2648 (765)* (P05038732!); 3119 (Calvinia): Namaqualand, river bed W of Brandkop (−AC), 9 December 1946, *F.M. Leighton 2441* (BOL!); 3119 (Calvinia): Zwart Doorn River, W of Brandkop (−AC), 9 December 1946, *R.H. Compton 18893* (BOL!; NBG! Excluding two annual plants); 3120 (Williston): An der Dualls Slangenfontein [Slangfontein] (−BD), (I, d1 d), 3000–4000 ft elev., 17 November 1826, *J.F. Drège* s.n. (P05038728!); 3120 (Williston): Sandwef [sic] on road to Brandvlei (−AC), 29 November 1986, *G. Germishuizen 4011* (NBG!); 3120 (Williston): Hantam, Kookfontein farm, Kookfontein rivier at crossing R-354 (−CA), 31°43′35″ S 20°14′07″ E, 1082 m elev., in saline substrate of ravine, 25 August 2022, *M. Martínez Azorín et al.* s.n. (ABH83290!); 3121 (Fraserburg): Karoo Region, near Fraserburg (−DC), 4200 ft elev., January 1888, *H. Bolus 10381* (NBG!); 3121 (Victoria West): Little Namaqualand, common on bed of Brakrivier (−BD), 1600 ft elev., 11 December 1908, *H.H.W. Pearson 4864* (K!); ibidem, *H.H.W. Pearson 4868* (BM!, K!); 3123 (Victoria West): Central Cape, Victoria West District, Hutchinson, Zeisiesfontein (−AC), 1260 m elev., *E.M. Nortje 10* (NBG!); 3220 (Sutherland): Tankwa Karoo, between Middlepos and Ganaga Pass (−CB), 32°37′46.7″ S 20°21′40.5″ E, 573 m elev., 26 August 2022, *M. Martínez Azorín et al.* s.n. (ABH83295!); 3221 (Merweville): Fraserburg, Ratelfontein vel “Balmoral” (−BA), 4500 ft elev., January 1888, *H. Bolus 10381* (BOL!, NBG!); 3221 (Merweville), Upper Region, Kopjies Kraal, river bed (−BA), 2000 ft elev., 12 December 1908, *H.H.W. Pearson 4886* (K!). Capland: Boschjemanskarroo [probably near Bitterfontein], 3000–4000 ft elev., November, *J.F. Drège* s.n. (HBG516896!). Boschjemans-karroo oder Onderbokkeveld, [3000–4000 ft elev., November], *J.F. Drège 6242* (P05038787!). Afrique austral, Herb. J. Hennecart, Drège, locum, 69 n° 1, [probably near Platberg], November 1838–1839, *J.F. Drège* (K!, P05038788!). **Western Cape Province**: 3118 (Vanrhynsdorp): Knersvlakte, Kalkgat farm (−BB), 255 m elev., 18 June 1987, *C. Boucher 5175* (NBG!); 3219 (Wuppertal): SW Cape Region, foot of Katbakkies Pass, east side, at Skitterykloof picnic site (−DC), 1800 ft elev., 6 January 1976, *H.C. Taylor 9049* (BM!, NBG!, K!); 3219 (Wuppertal): Karoo, Beukesfontein, sandy river bed (−CD), 1420 ft elev., 1 December 1908, *H.H.W. Pearson 5005* (BM!, BOL!, K!, P05038725!); 3219 (Wuppertal), Central Karoo District: Pappekuil [Papekuil] (−BC), not far from river, in sand, 950 ft elev., 3 November 1908, *H.H.W. Pearson 3985* (K!); 3219 (Wuppertal): Karoo, North of Gansfontein, river bed (−DA), 1200 ft elev., bush 1–1½ ft elev., 2 December 1908, *H.H.W. Pearson 3984* (K!); 3319 (Worcester): Worcester, near Mowers station (−DA), 10 November 1964, *Van Breda 1758* (NBG!); 3320 (Montagu): Laingsburg, Wittebergen near Matjiesfontein (−BA), October 1908, *R. Marloth 11442* (NBG!); 3321 (Ladismith): Riversdale Div., Klein Karroo, damp places in river beds (−CC?), 1200 ft elev., October 1924, *J. Muir 3546* (BOL!).

*Unidentified sites*: South Africa. Cap de Bonne Espérance, collection de Drège s.n. (HBG516896!, P05038791!). Cape, *Drège* s.n. (K!). Cap, 1838, *Drège* s.n. (P05144899!). Afr[ica]. Austr[alis]., 1836, *J.F. Drège* s.n., sub *F. nodiflora* (BM!, K!, P05038789!).

*Digital iNaturalist images*: South Africa. **Eastern Cape Province**. 3323 (Willowmore): Graaff-Reinet, Dr Beyers Naudé Local Municipality, Timbila Nature reserve, Grootrivier bed (−BB), 33°11′14″ S 23°53′16″ E, 580 m elev., 29 September 2019, *K. Jolliffe* s.n. (v.v.): https://www.inaturalist.org/observations/33692903; accessed on 3 April 2023.

#### 3.4.3. ***Frankenia anneliseae*** M.B.Crespo, M.Á.Alonso, Mart.-Azorín, J.L.Villar & Mucina, **sp. nov.**

*Type*: South Africa. Northern Cape: Namaqualand, Steinkopf, 11 December 1897, *Schlechter 40* (holo.: BOL!; iso.: BM!, K!, P05038802!).

-*F. capitata* auct. pl. atque *F. repens* auct. pl.


*Diagnosis: Planta speciosa a F. repenti caulibus valde lignosis et calycibus dense puberulis (interdum etiam costis hirsutis) remote accedenti, sed ab ea distinctissima et bene distinguenda caulibus erectis non radicantibus; petalis brevioribus 5.5–6.5 mm long. (non 9–11 mm); calycibus minoribus 3–5(–6) mm long. (non 6–8 mm); et praesertim seminum testa papillis brevioribus, 10–43 μm long, heterogeneis, aliis globosis aliis conico-obtusis (non papillis longioribus 55–110 μm, subhomogeneis, omnibus conico-obtusis).*


*Description*: *Shrub* densely branched, tap-rooted, woody at the base with grey bark, and glabrous to sparsely hairy. The *stems* are non-rooting, erect to ascending, fruticose, usually with erect, and with often divaricate branches 15–45 cm long; the *young branchlets* are with internodes up to 20 mm long, yellowish to reddish, ± densely pubescent (rarely glabrescent), covered all around with homogeneous indumentum of minute claviform or hooked trichomes (ca. 0.1–0.2 mm) and longer complanate trichomes (up to 0.4 mm long), denser below nodes, and somewhat retrorse to patent. The *leaves* are opposite-decussate, patent to erect-patent, deep-green, somewhat glaucescent, glabrous on the upper side and ± densely papillate beneath, and mostly with scattered whitish depositions; the *petiole* is 0.3–0.5 × 0.2–0.5 mm, flattened, and tapering distally; the *sheath* extends along the margins of the petiole almost to the blade, densely ciliate, with 6–10 pairs of cilia 0.4–1.2 mm long, unequal, filiform, and whitish; the *leaf blade* is 2–4 × 1–2 mm, broadly oblong to elliptic, rounded to subacute at apex, somewhat fleshy, often discolorous, abaxially paler with minute glands usually bearing salt depositions, with margins often strongly revolute and hiding the abaxial side almost completely, and somewhat thickened; the *midrib* is thickened, tapering slightly towards the apex, continuous with the petiole and raised abaxially, and extending about half the blade length; the *young leaves* are on short shoots, fasciculate and similar but smaller and sometimes narrower than those on long shoots. The *flowers* are pentamerous, perfect, borne commonly in dichasial groups, usually branched, with erect branchlets up to 15 mm long, but often crowded in compact inflorescences; the *lowermost flowers* are in reduced groups and briefly pedunculate, and the uppermost dichasia are often denser and subcorymbose (the central flower always sessile, the lateral ones on pedicels 0.5–1 mm long). The *floral bracts* are two in number, 2–3.5 mm long, leaf-like, erect-patent to erect, connate at base, and enveloping basal part of the calyx for ca. 0.5 mm. The *bracteoles* are two in number, 1–2 mm long, bract-like but smaller, about half the length of calyx, adnate to the calyx base, and alternating with bracts. The *calyx* is 3–5(–6) × 0.8–1.5 mm, fusiform-tubular to gradually fusiform after anthesis, untwisted, straight, indurate, with 5 prominent thickened ribs, densely papillate (papillae whitish, minute, claviform, or globose) between the ribs but sparsely hirtellous on the ribs (trichomes whitish, up to 0.2 mm long), and sometimes with scattered whitish depositions on the upper part; the *teeth* are five in number, 0.8–1.2 mm long, triangular, acute or briefly mucronulate (mucro ca. 0.2 mm long), minutely papillate, often yellowish, and not recurved after anthesis. The *petals* are five in number, 5.5–8 × 0.9–1.5 mm, long obovate-cuneate, and whitish to pinkish-mauve or purplish; the *claw* is 2–3 × 0.6–0.7 mm, narrowly cuneate, imperceptibly tapering to blade, whitish, and hidden into the calyx tube; the *ligule* is 1.5–2.5 × 0.3–0.4 mm, narrowly oblanceolate, longitudinally adnate to claw, the free apex ca. 0.5 mm long, ovate-acute to acuminate, and entire to slightly denticulate on margins; the *blade* is 2.3–3.5 × 1.5–2 mm, broadly obovate to suborbicular with rounded to truncate, and irregularly erose-denticulate apex. The *stamens* are six in number, in two unequal whorls, long exserted, and overtopping 1.5–2.5 mm the calyx teeth at anthesis; the *filaments* are 6–8 mm long, expanded ca. 0.5 mm wide in the lower half but gradually tapering and filiform in the distal half, and pinkish to bluish-pink; the *anthers* are 0.4–0.6 mm long, ellipsoid, versatile, and yellow. The *ovary* is ellipsoid, subtrigonous, and with three carpels; the *placentae* are three in number, parietal-basal, extending up to the lower half to two thirds of carpel wall length, and with ventral traces moderately to highly branched; the *ovules* are 5–6 per placenta and attached along most of the placenta by erect funiculi 0.2–0.4 mm long. The *style* is 7–8 mm long, terete, somewhat sigmoid at the base, exserted and elongating up to 11 mm after pollination, and whitish; the *style branches* are three in number, filiform, 0.9–1.5 mm long, and whitish; the *stigmas* are slightly clavate. The *capsule* is 2.5–3.5 × 1–2 mm, ovoid-ellipsoid, hidden in the calyx tube, dark reddish-brown, and early dehiscent. The *seeds* are 1–8 per capsule, 0.7–1.1 × 0.3–0.4 mm, ellipsoid to ovoid-ellipsoid, sulcate on one side, pale brown, darker at the funicular part, and developing rapidly even before the flower has completely withered; the *testa* is thin, not mucilaginous, with surface weakly and irregularly ornamented with a subrectangular-reticulate pattern, finely striate, sparsely covered with medium-sized papillae 10–43 μm long, heterogeneous, globose to conical-obtuse, and more densely disposed on the distal part.

*Etymology*: The specific epithet (*Annelisea, –ae* = belonging to Annelise) honours Annelise le Roux for her enormous contribution to the knowledge of the flora of Namaqualand, and gigantic conservation efforts protecting it. Annelise made us aware of this taxon and identified it as possibly a new species native to that unique semi-desert region of NW South Africa.

*Phenology*: The species flowers in late October–early January (occasionally in July–August) and fruits occur in November–February (occasionally in August–September).

*Habitat and distribution*: *Frankenia anneliseae* grows in saline, well-drained sandy, azonal soils of salt pans, and beds of intermittently flowing (occasionally during regional precipitation-rich events) semi-desert rivers, ravines, and streams in inland regions at elevations of 300–1000 m above sea level (Figure 7). The known distribution of the species is restricted to the NW part of the Karoo Region in NW South Africa, ranging from Eksteenfontein and Steinkopf to Klipfontein in the Northern Cape Province (Figure 4), a territory included in the northern Succulent-Karoo (SK) biomes (mostly in the SKn, SKr, and SKs bioregions) *sensu* Mucina and Rutherford [14], where it specifically inhabits the so-called “Namaqualand Salt Pans” (code AZi 2). In those areas, the climate is subdesert to arid and continental (not or scarcely ameliorated by the ocean influence), with average temperatures ranging from 5 °C in winter to 30 °C (or even more) in summer and frosts being absent or scarce, but much varies between years. The average annual precipitation ranges around 70–200 mm, though rather differently distributed and with occasional local rains reaching about 300 mm; the rainfall occurs mostly during winter (May to September) with a peak in June and episodic drought periods well below 100 mm per year being frequent [14].

*Notes*: Wild populations of *F. anneliseae* include numerous individuals covering a large territory in NW South Africa, and no special threats are known so far to occur that might lead to any inferred decline in either the number of populations or the number of individuals. Therefore, its conservation status is suggested here as Least Concern (LC) according to IUCN [42].

*Studied material:* South Africa. **Northern Cape Province**: 2817 (Vioolsdrift): Skoverfontein, ca. 12 km NW of Eksteenfontein (−CC), P15, 28°45′50″ S 17°09′15″ E, 448 m elev., 19 August 2022, *M. Martínez Azorín et al.* s.n. (ABH831912!); 2817 (Vioolsdrift): Skoverfontein, ca. 12 km NW of Eksteenfontein (−CC), P16, 28°46′25″ S 17°09′41″ E, 460 m elev., 19 August 2022, *M. Martínez Azorín et al.* s.n. (ABH83196!); 2817 (Vioolsdrift): Little Namaqualand: Stinkfontein, river bed (−CD), 5 December 1910, *H.H.W. Pearson 5526* (K!); ibidem, near Stinkfontein, dry river bed, 25 December 1910, *H.H.W. Pearson 5967* (K!); 2817 (Vioolsdrift): Skoverfontein, ca. 12 km NW of Eksteenfontein (−CC), P17, 28°46′33″ S 17°10′18″ E, 481 m elev., 19 August 2022, *M. Martínez Azorín et al.* s.n. (ABH83198!); 2917 (Springbok): Namaqualand, Steinkopf (−BC), 11 December 1897, *Schlechter 40* (BM!, BOL!, P05038802!); Namaland Minor [near Springbok], *W.C. Scully 9* (BM!, BOL!, P05038765!); 3017 (Hondeklipbaai): Klipfontein, 2–3 km N of Klipfontein, S of Kersboshoek (−BD), 30°28′43.3″ S 17°49′40.9″ E, 309 m elev., 30 August 2017, *M. Martínez Azorín et al.* s.n. (ABH76891!); ibidem, Klipfontein, open hillside, but near water course (−BD), 20 December 1949, *E.C. Macdonald 109* (BM!); ibidem, Klipfontein, bottom of valley of dry watercourse (−BD), 3000 ft elev., 23 December 1949, *E.C. Macdonald 118* (BM!); 3018 (Kamiesberg): Namaqualand, Kamabies (−CB), on side (stony) of dam, 3000 ft elev., 24 December 1908, *H.H.W. Pearson 3462* (K!); 3018 (Kamiesberg): Namaqualand, Kamabies (−CB), dry sandy river bed, 24 December 1908, *H.H.W. Pearson 3955* (BM!, K!);.3018 (Kamiesberg): Upper Bushmanland, Nieuwefontein (−DA), 2700 ft elev., 20 December 1908, *H.H.W. Pearson 3467* (K!).

*Unidentified sites*: South Africa. Namaqualand, *W.C. Scully 226* (BM!).

*Digital iNaturalist images:* South Africa. **Northern Cape Province**. 2817 (Vioolsdrif): Namaqualand, Eksteenfontein, Sendelingsdrif (−CC), 28°46′29″ S 17°09′45″ E, 458 m elev., wadi [sic] bed, 12 October 2018, *S. Swanepoel* s.n. (v.v.): https://www.inaturalist.org/observations/17553648; accessed on 3 April 2023; 2917 (Springbok): Namaqualand, Komaggas area, NW Oubeep (−DC), 29°51′41.36″ S 17°33′53.26″ E, 581 m elev., river bed, 16 March 2017, *N. Helme* s.n. (v.v.): https://www.inaturalist.org/observations/11292132; accessed on 3 April 2023.

## 4. Discussion

### 4.1. Morphological and Evolutionary Relationships

The findings presented in this paper are part of an ongoing broader study evaluating the global generic and specific relationships in *Frankeniaceae* currently focused on the Eurasian and African taxa. Our ITS phylogeny, as shown in Figure 2, is still partial but, in combination with our morphological studies (summarised in Table 3) and field observations, brings a better scenario for a more accurate interpretation of the diversity of *Frankenia* in Southern Africa.

The three main groups (clades A–C) obtained with the studied Mediterranean and African (including Macaronesia) taxa are apparently not clearly supported by the gross morphology, perhaps due to the fact that convergent evolution in saline ecosystems might model similar morphological traits in members of phylogenetically distant lineages [43]. However, individual species discrimination is feasible based on characters such as habit, indumentum, leaf morphology, inflorescence structure, petal and calyx size or colour, or seed features, among others [39,41,43,44,45,46].

In the case of the *Frankenia repens* group (clade A), the internal molecular relationships among the three lineages found are rather well resolved and also well-characterised morphologically and biogeographically, and hence, they are treated here at species rank: *F. nummularia*, *F. anneliseae*, and *F. repens*.

All those species share a suffruticose to woody perennial habit, with flowers often disposed in dichasial and more or less condensed inflorescences; with an untwisted calyx; with an ovary with 15–40 parietal ovules; and with large seeds that are 0.7–1.3 mm long. However, important differences exist concerning particular seed features, which are diagnostic to support a distinction between species, although they apparently had never been studied in detail in the genus. First, *F. nummularia* is recovered in our trees (Figure 2) as strongly sister (1.00 PP, 90 BP) to the pair *F. repens*–*F. anneliseae*, from which it differs on account of its long petiolate (0.7–1.5 mm) leaves, which are mostly suborbicular or broadly elliptic, rounded to slightly pointed at apex, often flattened or longitudinally folded, and with margins that are loosely and gradually revolute; stems with internodes up to 30 mm long, which are much longer than leaves; a calyx that is 4.5–6 mm long with acute to minutely mucronate (mucro up to 0.2 mm long) teeth; a seed testa that is almost smooth, with small papillae at base only that are 3.5–9 μm long, homogeneous, and conical-obtuse in shape (Figure 3e,f). Second, *F. anneliseae* is weakly supported as sister (0.60 PP, 60 BP) to *F. repens* in our trees, as both are superficially closer due to its shortly petiolate (0.3–0.5 mm) leaves, with the blade that is strongly revolute on the margins to hide the abaxial side almost completely; and seeds covered with much larger papillae, which are up to 110 μm long (Figure 3g,h,i,j). All this led to a misidentification of *F. anneliseae* with *F. repens* or its synonyms [47,48] or also with the Macaronesian endemic *F. capitata* [38]. However, the weak sister support of both lineages is congruent with their sound morphological differences, *F. anneliseae* clearly differing from the typical *F. repens* (lectotype: SBT10193 [digital image!]) based on (Table 3):(i)ascendent to erect stems (vs. prostrate, usually rooting stems);(ii)a smaller (3–6 mm vs. 6–8 mm) calyx;(iii)a distinct petal length (5.5–8 mm vs. 9–11 mm); and(iv)a seed testa with smaller papillae (10–43 μm vs. 55–110 μm long), that are heterogeneous in shape, globose to conical-obtuse (vs. subhomogeneous in shape, cylindrical-conical obtuse), and that are sparsely disposed but denser on the distal part (vs. more densely disposed all over the testa surface; Figure 3g,h).

Furthermore, although no useful sequences were obtained for *F. nodiflora,* and hence, its phylogenetic relationships remain unknown, its morphological traits are distinct enough (Table 3) for an unequivocal and easy differentiation based on:(i)concolorous leaves, which are bright green on both sides (or slightly paler beneath);(ii)inflorescences that are mostly axillary and glomerular and borne on short lateral opposite branches;(iii)calyx teeth that are conspicuously acuminate with acumen up to 0.7 mm long;(iv)an ovary with 4–6 ovules per placenta; and(v)a seed testa that is sparsely covered with medium-sized papillae that are 12–33 μm long, heterogeneous, and globose to conical-obtuse (Figure 3c,d).

Accordingly, the name *F. nodiflora* is restored here as initially described by Lamarck [13] and applied by South African botanists in the first half of the 19th century [47]. Some superficial resemblance of both *F. nodiflora* and *F. nummularia* to the annual or short-lived perennial *F. pulverulenta*, such as the broad, flattened or slightly revolute on the margins, long-petiolate, and almost glabrous leaves and the relatively small flowers, usually led to a misidentification and consequently to synonymisation to the latter. However, the typical individuals of *F. pulverulenta* (lectotype: LINN 457.6!) can be easily distinguished (vs. both *F. nodiflora* and *F. nummularia*) by many taxonomically relevant characters (Table 3), such as:(i)a slender annual habit (vs. suffruticose perennial) with stems and branches that are pubescent on only one side with minute curled hairs (vs. glabrous or covered with minute curled and long flexuous hairs up to 1.2 mm long);(ii)leaf blade that is long and hairy on the abaxial side (vs. glabrous on both sides);(iii)flowers that are solitary and scattered along the branches (vs. crowded in loose or dense terminal and/or axillary dichasia);(iv)petals that are 3.5–5 mm long (vs. 5–9 mm);(v)a calyx that is 2.5–4(5) mm, puberulous on groves or glabrous (vs. entirely glabrous to sparsely puberulous to hirtellous at base and on ribs), with two bracteoles oblong-linear, as long as or longer than the calyx tube (vs. suborbicular to broadly ovate-oblong, much shorter than the calyx tube); and(vi)seeds that are very numerous, up to 45 per capsule, ca. 0.5–0.7 mm long (vs. less numerous, up to 22 per capsule, ca. 0.7–1.1 mm long), with testa very sparsely covered with small papillae that are 4–17 μm long, homogeneous, and conical-obtuse (Figure 3a,b).

Obermeyer [4] and other previous authors regarded *F. nodiflora* (treated in a very broad sense to also include *F. nummularia* and *F. nodiflora*) as a perennial form of *F. pulverulenta* occurring in the subdesert parts of South Africa: “[recorded] further inland in the drier areas, in saline surroundings such as salt pans and banks of brackish streams. /…/ Collections indicate that the plants may behave as annuals or occasionally persist as perennials when they become more woody and larger”. In this respect, annual plants belonging to *F. pulverulenta* s.l. can grow together with other perennial members of the genus, and sometimes, they have been collected in a single site. In fact, vouchers *R.H. Compton 18893* (NBG!) and *R.H. Compton 18894* (NBG!), both identified as *F: pulverulenta* and collected in Zwart Doorn River, W of Brandkop (Calvinia) on 9 December 1946, are good examples illustrating this matter. The former (*R.H. Compton 18893*) includes several fragments of *F. nummularia* plus two entire annual individuals (affixed on the upper-left side and the lower-right corner of the voucher) with leaves that are pubescent on the abaxial surface and bracteoles equalling the calyx length (P. Winter pers. comm.) clearly belonging to the true *F. pulverulenta*, whereas the latter (*R.H. Compton 18894*) bears several individuals of the typical *F. pulverulenta*. Similarly, perennial plants of *F. nodiflora* co-occur with annual plants of *F. pulverulenta* in the surroundings of Cape Town, as deduced from a single herbarium sheet at K (“Herbarium Zeyheri—Cape, *Dr. Pappe*”), which includes three smaller herbaceous fragments matching *F. pulverulenta* affixed on the upper part of the sheet and marked “(1)” in pencil (barcode number K000232048!) plus four larger fragments of *F. nodiflora* (identified as “*Frankenia krebsii* ? /…/ an a Frank. capitata Webb diversa? /…/ non Frankenia nothria Thbg,”) in the central and lower parts of the sheet (barcode number K000232047!) and marked “(2)” in pencil.

Most likely, the occurrence of both annual and perennial individuals sharing some morphological characters might have led researchers to mistakenly consider them as conspecific under the name *F. pulverulenta* and, hence, neglect the existence of other well-characterised entities deserving of taxonomic recognition. In fact, our trees (Figure 2) show that diverse samples of the typical *F. pulverulenta* (from distant territories of South Africa, the Canaries and the Western Mediterranean basin) form a compact group in clade B that is far apart from the sequenced South African members of the “*F. repens* group” in clade A. According to the morphological and biogeographical affinities of *F. nodiflora*, this neglected South African endemic would be expected to group together with the remaining species related to *F. repens*. Our effort is currently focussed on trying to obtain new samples of *F. nodiflora* to test that hypothesis.

Regarding other woody perennial Southern African taxa, such as *F. fruticosa* and *F. pomonensis*, both can easily be separated by exclusive combinations of characteristics not found in any of the newly described species. First, *F. fruticosa* (holotype: *N.A. Helme 7796*, NBG!) is a gnarled erect shrub with thickened stems and is strongly woody at the base (many-branched); the leaves are allantoid, small (1.5–2.5 × 0.5–1 mm), and densely puberulent on both sides; the flowers show petals with claws that are imperceptibly tapering and scarcely narrower than the blade; and stamens are long exserted after anthesis and contain only six ovules per ovary (one pair near the base of each of three subbasal-parietal placentas). These distinctive morphological traits make *F. fruticosa* a unique species in Africa [10], which should be regarded as an edaphic specialist that evolved after ecological diversification and adaptation to the peculiar conditions provided by quartz patches, to which it is endemic in Knersvlakte Bioregion, western South Africa [49]. Its position in all our phylogenetic trees as a sister to the remaining studied taxa (100 BP in the MP tree; see Appendix A) accords with its outstanding and unique morphological features in the context of the African and Eurasian members of *Frankenia*. Our ongoing research is also focused on investigating the relationships of this species with the Australian and American lineages of the genus. Second, *F. pomonensis* (holotype: M0104482 [digital image!]) differs by being a sprawling shrublet with procumbent to ascendent stems up to 1 m tall; leaves that are 3–6 mm long and minutely puberulous or papillate beneath; flowers that are scattered and in loose terminal dichasia; a calyx that is 5.5–6.5 mm long and glabrous; petals that are 8–9 mm long and purplish; and stamens that are much shorter than the petals and scarcely protruding from the corolla throat [50]. This remarkable plant is only known from the coastal areas of the Namib desert in southern Namibia, where it occurs on brackish flats, dolomite outcrops facing the sea, and bordering saline lagoons [4]. So far, no molecular and phylogenetic data are available for *F. pomonensis*.

According to our current knowledge, given the molecular and morphological distinctiveness and their non-overlapping distribution areas, we suggest recognition of all the studied South African entities in the *F. repens* group at species rank: *F. nodiflora*, *F. nummularia*, *F. anneliseae*, and *F. repens*. Future molecular work will shed light on the phylogenetic relationships of *F. nodiflora* with the remaining Southern African taxa of the genus. With the newly described and restored species, *Frankenia* is constituted by seven species in South Africa, which can be identified using the dichotomous key shown below.

Further research, including field observations and molecular work, is being carried out specifically on coastal populations of the *F. repens* group to clarify their taxonomic adscription. Similarly, the annual or short-lived perennials resembling *F. pulverulenta*, which occur in the subdesert and desert areas of both northern and southern Africa, are currently being evaluated to better determine its identity and relationships.

### 4.2. Phytogeographic and Ecological Patterns

The geographic distribution of the *Frankenia* taxa recognised in this paper matches patterns found earlier in other genera occurring in two neighbouring and bioclimatically contrasting semi-desert biomes of Southern Africa—the Succulent Karoo (characterised by winter-rainfall) and the Nama-Karoo (characterised by bimodal precipitation regime; see [51,52,53]). The most prominent examples are found in *Heliophila* Burm.f. ex L. [54], *Caroxylon* Thunb. [55], *Austronea* Mart.-Azorín & al. [56], or *Spergularia* (Pers.) J. Presl and C. Presl [57], among many others. Equally well is documented the edaphic specialisation characteristically occurring in clades limited to one of the biomes; see, for instance, the recent cases of *Triglochin* Riv. ex L. [58], *Ursinia* Gaertn. [59], *Sarcocornia* A.J.Scott [60], *Limonium* Mill. [61], *Spergularia* [41], or *Cotula* L. [62]. We suggest that these intrageneric patterns are the results of recent rapid radiations [63], which took place in the Plio–Pleistocene period following successive disruptive aridification events that enhanced the establishment of taxa in habitats often of contrasting ecology, including the salt-laden ones [64].

## 5. Identification Key for Southern African Species of *Frankenia*

1.Leaf blade broad, suborbicular to broadly oblong or obovate, flat or longitudinally folded, and sometimes slightly recurved on margins and on the distal half; petiole distinct and 0.5–1.5 mm long......................................................................................................................................................................................2

-Leaf blade linear to linear-lanceolate, mostly strongly recurved on margins, allantoid, or sometimes somewhat flattened; petiole inconspicuous and up to 0.5 mm long ......................................................................................................................................................................................4

2.Plants annual, delicate. Leaves and bracts hairy beneath. Flowers scattered and solitary in the forks of the branches. Bracteoles as long as or longer than the calyx. Petals 3.5–5 mm long. Seeds up to 45 per capsule, small, 0.5–0.7 mm long................................................................................................................................***F. pulverulenta***

-Plants suffruticose perennial and shrubby. Leaves and bracts glabrous on both sides. Flowers in terminal dichasia. Bracteoles about half the length of the calyx. Petals 6–9 mm long. Seeds up to 22 per capsule, large, 0.7–1.3 mm long.................................................................................................................................................................3

3.Leaves of flowering stems broadly ovate and subacute. Inflorescence mostly glomerular and condensed at stem nodes. Calyx 4–4.5 mm long; teeth with a long, conspicuous acumen up to 0.7 mm long. Seed testa sparsely covered with medium-sized papillae 12–33 μm long, heterogeneous, globose to conical-obtuse....................................................................................................................................................................................***F. nodiflora***

-Leaves of flowering stems suborbicular to elliptic and obtuse. Inflorescence loosely dichasial and usually widely branched. Calyx 4.5–6 mm long; teeth acute or with an inconspicuous mucro up to 0.2 mm long. Seed testa almost smooth with small papillae 3.5–9 μm long only at the base, homogeneous, conical-obtuse....................................................................................................................................................................................***F. nummularia***

4.Stems stout, tortuous, and strongly woody at the base. Leaves 1.5–2.5 mm long. Calyx ribs densely puberulous. Petal blade scarcely wider than the claw. Ovules 6, subbasal-parietal....................................................................................................................................................................................***F. fruticosa***

-Stems thin and suffruticose. Leaves 3–7 mm long. Calyx ribs glabrous or sparsely hairy. Petal blade distinctly wider than the claw. Ovules 15–40, parietal....................................................................................................................................................................................5

5.Plant sprawling, up to 1 m tall. Stems glabrous. Calyx glabrous. Stamens much shorter than the petals and scarcely protruding from the corolla throat....................................................................................................................................................................................***F. pomonensis***

-Plant compact, erect to prostrate perennial, and up to 45 cm tall. Stems variously indumented. Calyx puberulent in grooves and often with coarse whitish hairs on the ribs, rarely glabrous. Stamens protruding far from the corolla throat.................................................................................................................................................................6

6.Stems erect or ascendant, non-rooting, and covered with minute hooked trichomes and longer complanate trichomes. Petals 5.5–8 mm long. Calyx 3–5(–6) mm long. Seeds sparsely covered with medium-sized papillae 10–43 μm long, heterogeneous in shape, globose to conical-obtuse...................................................................................***F. anneliseae***

-Stems prostrate, often rooting at nodes, and covered with short curly hairs. Petals 9–11 mm long. Calyx 6–8 mm long. Seeds densely covered with large papillae 55–110 μm long, subhomogeneous in shape, cylindrical-conical obtuse............................................................................................................................................................***F. repens***

## Figures and Tables

**Figure 1 plants-12-02630-f001:**
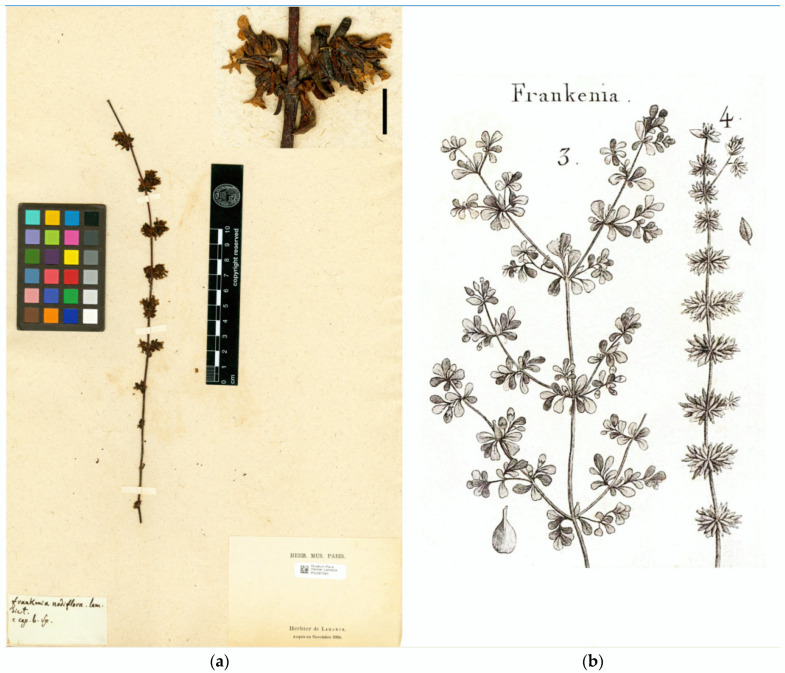
Relevant material of *F. nodiflora* Lam. (**a**) Lectotype here designated from Lamarck’s collection (P00287094!) with a close-up detail of inflorescences (scale bar: 5 mm); reproduced with permission (© Muséum National d’Histoire Naturelle, Herbarium, Paris); (**b**) Comparative illustrations of *F. pulverulenta* (left; num. 3) and *F. nodiflora* (right; num. 4), according to *Tableau encyclopédique et méthodique* of Lamarck [36], plate 262 [partially modified]).

**Figure 2 plants-12-02630-f002:**
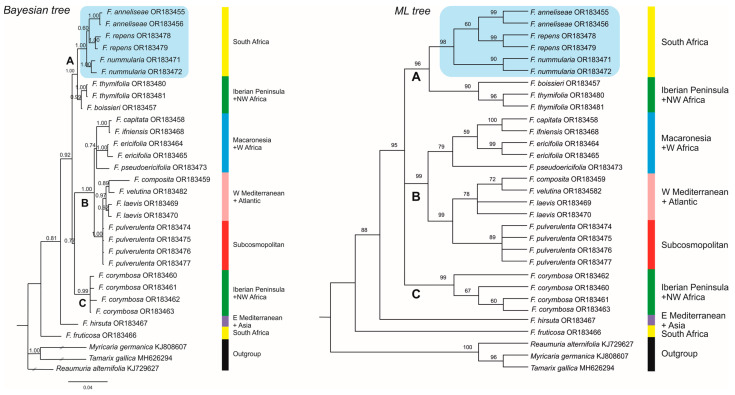
Bayesian Inference (BI) and Maximum Likelihood (ML) phylogenetic trees of *Frankenia* accessions from ITS nuclear DNA sequences. The three main clades recovered in the analyses are marked A, B and C in the trees. Members of the “*Frankenia repens* group” are highlighted in clade A. Numbers above branches indicate posterior probabilities (PP) from the Bayesian analysis in the BI tree (left), whereas they indicate the percentage of trees in which the associated taxa clustered together in the ML tree (right) after 10,000 bootstrap replicates. GenBank codes are shown after each taxon/accession name.

**Figure 3 plants-12-02630-f003:**
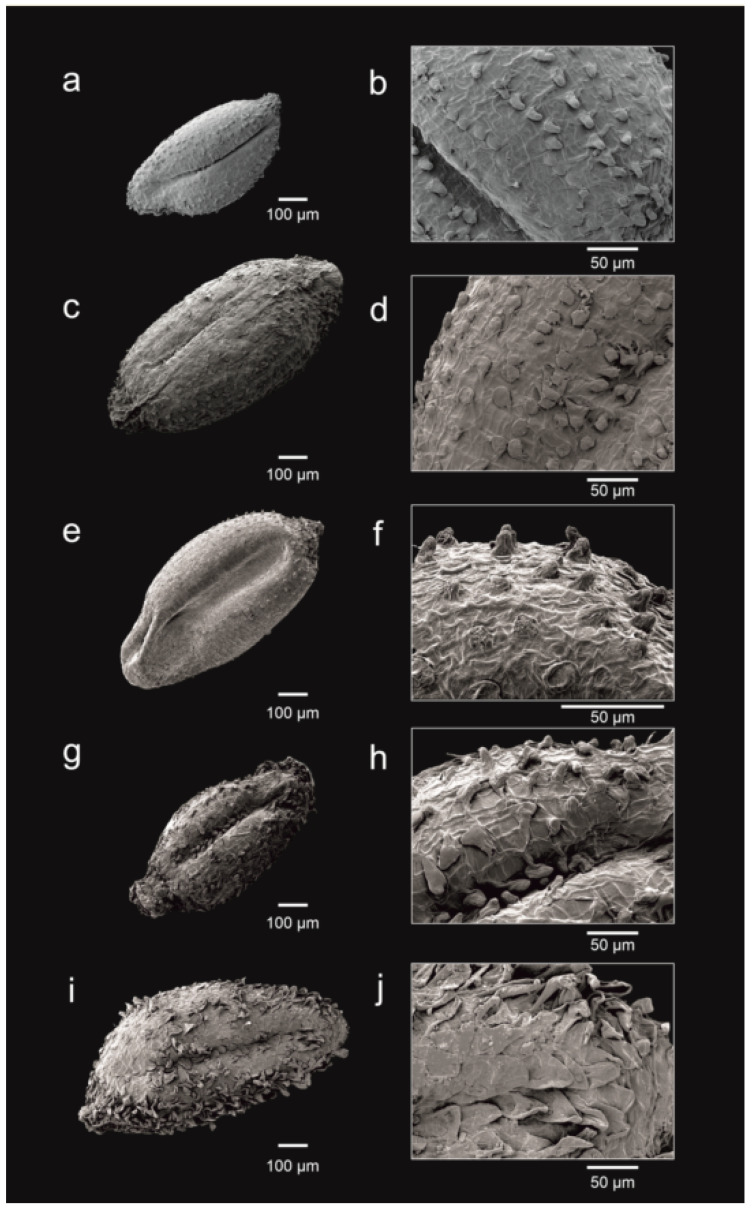
Seed morphology (left) and testa details (right) of: (**a**,**b**). *Frankenia pulverulenta* (ABH41888); (**c**,**d**). *F. nodiflora* (P05038804); (**e**,**f**). *F. nummularia* (ABH83290); (**g**,**h**)*. F. anneliseae* (ABH76891); and (**i**,**j**). *F. repens* s.l. (ABH76882).

**Figure 4 plants-12-02630-f004:**
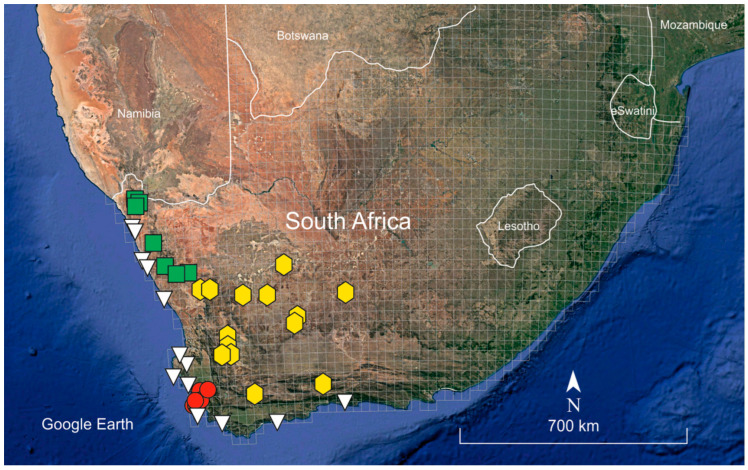
Distribution map of the studied material (both herbarium vouchers and iNaturalist data) of the “*Frankenia repens* group” in South Africa: *F. nodiflora* (red circles), *F. nummularia* (yellow hexagons), *F. anneliseae* (green squares), and *F. repens* (white triangles; see Appendix B for data source).

**Figure 5 plants-12-02630-f005:**
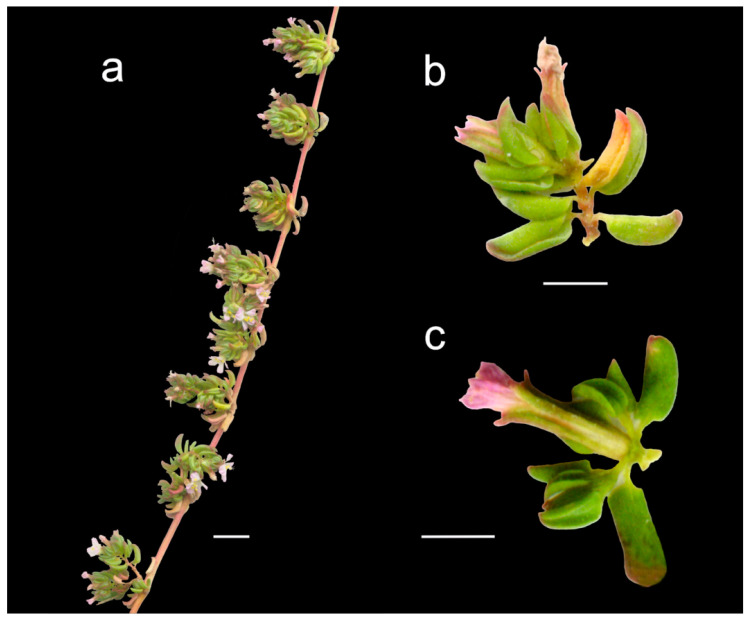
*Frankenia nodiflora* Lam. from Uitkamp N.R., Durbanville, Cape Town. (**a**) Fragment of stem with flowering branchlets; (**b**) Branchlet apex with details of leaves and inflorescence; (**c**) Detail of calyx at anthesis, with acuminate teeth (images: Hedi Stummer, 7 November 2014). Scale bars = 4 mm (**a**), 2 mm (**b**,**c**).

**Figure 6 plants-12-02630-f006:**
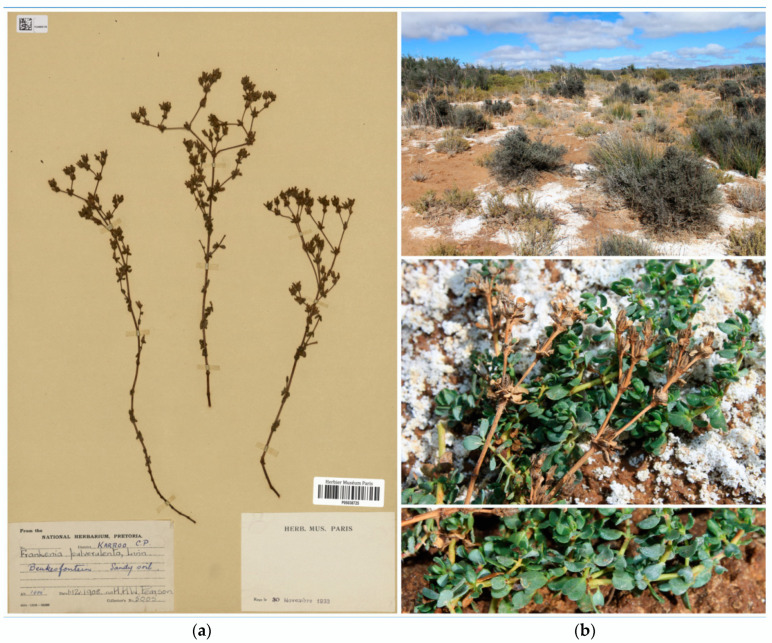
*Frankenia nummularia* sp. nov. (**a**) Western Cape: Beukesfontein, *H.H.W. Pearson 5005*; isotype: P05038725); available at: https://science.mnhn.fr/institution/mnhn/collection/p/item/p05038725; accessed on 3 April 2023 (© Muséum National d’Histoire Naturelle, Herbarium, Paris); (**b**) Northern Cape: Kookfontein river; living plants in habitat, with details of leaves and withered inflorescences.

**Figure 7 plants-12-02630-f007:**
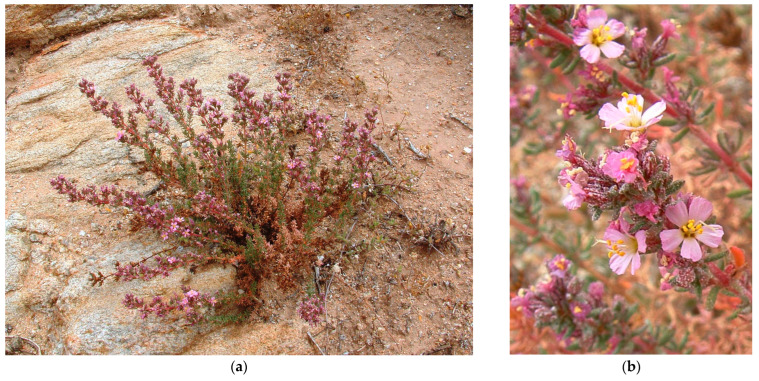
*Frankenia anneliseae* sp. nov. (**a**) Plant in habitat, near Springbok (Northern Cape Province); (**b**) Details of the inflorescences at anthesis (photos: Veronica Esterhuizen, 1 November 2014).

**Table 1 plants-12-02630-t001:** Studied seed samples of *Frankenia* with provenance and herbarium vouchers.

Taxon	Locality	Herbarium Voucher
*F. anneliseae*	South Africa: Steinkopf	P05038802
	South Africa: Klipfontein	ABH76891
	South Africa: Skoverfontein	ABH83198
*F. nodiflora*	South Africa: Cape Town	P05038804
	South Africa: Uitkamp	ABH83529
	South Africa: C.T., Salt River	K s.n.
*F. nummularia*	South Africa: Beukesfontein	P05038725
	South Africa: Kookfontein River	ABH83290
	South Africa: Gansfontein	K s.n.
*F. pulverulenta*	South Africa: Redelinghuys	ABH77205
	Spain: Alicante, Jávea	ABH41888
	Spain: Tenerife, Puerto de la Cruz	ABH79974
*F. repens*	South Africa: Port Nolloth	ABH76882
	South Africa: S of Groenrivier	ABH76868
	South Africa: S of Hondeklipbaai	ABH76862

**Table 3 plants-12-02630-t003:** Comparison of morphological characters of *Frankenia nodiflora* to other morphologically related species from South Africa.

	*F. nodiflora*	*F. nummularia*	*F. anneliseae*	*F. repens* s.l.	*F. pulverulenta* s.l.
General habit	shrublet, densely branched	shrublet, densely branched	shrub, densely branched	shrublets, lignified at base	annual or rarely short-lived perennial
Stem features	rooting only at base, diffuse, creeping prostrate	rooting only at base, procumbent to ascending	erect, densely disposed	mostly rooting at nodes	prostrate to ascending, non-rooting
Branchlet indumentum	glabrous to shortly and loosely puberulous all around	glabrous to ± densely hairy mostly on one side	±densely pubescent all around (rarely glabrescent),	± densely pubescent all around	±densely pubescent on one side
Branch trichomes: types and length	minute and scattered, up to 0.05 mm, or absent	curled, ca. 0.2 mm, plus flexuous, 0.7–1 mm	claviform or hooked, ca. 0.1–0.2 mm, plus complanate, up to 0.4 mm	minute, curled or hooked, ca. 0.1–0.2 mm	minute, curled, ca. 0.1–0.2 mm
Petiole length × width (mm)	0.6–1.2 × 0.2–0.3	0.7–1.5 × 0.3–0.5	0.3–0.5 × 0.2–0.5	0.5–1.5 × 0.2–0.3	0.5–1.5 × 0.2–0.3
Petiole sheath	2–5 pairs of cilia to subglabrous	2–5 pairs of cilia	6–10 pairs of cilia	3–4 pairs of cilia	2–6 pairs of cilia
Sheath cilia length (mm) and colour	0.2–0.8, unequal, whitish	0.3–0.8, unequal, whitish	0.4–1.2, unequal, whitish	0.8–2, unequal, whitish	0.3–0.6, unequal, whitish
Leaf blade length × width (mm)	2.5–3.5 × 1–2	2–7 × 1.5–6	2–4 × 1–2	4–7 × 1–1.5	2–5 × 1.5–3
Leaf blade outline and colour	broadly ovate-elliptic to oblong, concolorous (green)	suborbicular or broadly elliptic, often discolorous	broadly oblong to elliptic, often discolorous	narrowly linear, often discolorous	broadly obovate-cuneate, subconcolorous (greyish-green)
Leaf blade apex and margins	subacute to mucronulate; gradually revolute	rounded to slightly pointed at apex, loosely and gradually revolute	rounded to subacute at apex, often strongly revolute	subacute at apex, strongly revolute, allantoid	rounded, slightly emarginate, flattened or slightly revolute
Leaf blade indumentum	absent (glabrous) on both sides	absent (glabrous) on both sides	absent (glabrous) on both sides	absent (glabrous) or pubescent above, ± densely pubescent beneath	absent (glabrous) above, ± densely pubescent beneath
Inflorescence	mostly glomerular, condensed at stem nodes	loosely dichasial, usually widely branched	dense dichasial, usually branched, but compact	dense dichasial, usually branched, but compact	flowers solitary and scattered along branch dichotomies
Bracteole length (mm)	1–2, about half to 2/3 of calyx	1–2, up to half of calyx	1–2, about half of calyx	2–4, about half to 2/3 of calyx	2.5–4, as long as or longer than calyx
Calyx length × width (mm), shape and torsion	4–4.5 × 1–1.5, tubular, untwisted	4.5–6 × 1–1.5, fusiform-tubular to fusiform, untwisted	3–5(–6) × 0.8–1.5, fusiform-tubular to fusiform, untwisted	6–8 × 1–2, fusiform-tubular to fusiform, untwisted	2.5–4(–5) × 0.8–1.5, fusiform-tubular to fusiform, untwisted
Sepal indumentum (appearance and length)	absent (glabrous)	absent (glabrous) or sparsely puberulous to hirtellous (trichomes up to 0.5 mm)	minutely papillate between ribs, sparsely hirtellous on ribs (trichomes up to 0.2 mm)	absent (glabrous) or puberulous to hirtellous (trichomes up to 2 mm)	absent (glabrous) or minutely papillate between ribs, (trichomes up to 0.2 mm)
Calyx teeth length (mm)	0.9–1.2, with subapical acumen ca. 0.7 mm	0.7–0.9, acute to mucronate (mucro up to 0.2 mm)	0.8–1.2, acute to mucronate (mucro ca. 0.2 mm)	1.2–2, acute to mucronate (mucro ca. 0.2 mm)	0.4–0.8, acute to mucronate (mucro ca. 0.1 mm)
Petal size (mm) and colour	5–7 × 1–2, pinkish-mauve but whitish below	6–9 × 1–2, pinkish-mauve	5.5–8 × 0.9–1.5, whitish to pinkish-mauve or purplish	9–11 × 2–3.5, pinkish-mauve to purplish	3.5–5 × 0.6–0.9, whitish-pink to pinkish-mauve
Petal blade size (mm), and shape	2.3–3 × 1.5–2, broadly obovate, rounded to truncate, slightly emarginate, ±sinuate apex	2.3–4 × 1.5–2, broadly obovate to suborbicular, rounded to truncate, erose-denticulate apex	2.3–3.5 × 1.5–2, broadly obovate to suborbicular with rounded to truncate, erose-denticulate apex	4–5 × 2.5–3.5, broadly obovate to suborbicular with rounded, erose-denticulate apex	2–2.5 × 0.5–0,8, narrowly cuneate to obovate, with truncate, erose-denticulate apex
Petal claw (mm)	2–3 × 0.5–0.7, narrowly cuneate	4–5 × 0.8–1, narrowly cuneate	2–3 × 0.6–0.7, narrowly cuneate	5–6 × 1.5–2, narrowly cuneate	2–2.5 × 0.3–0.5, cuneate
Petal ligule (mm)	1.5–2.5 × 0.3–0.4, free apex ca. 0.5 mm, oblong-obtuse, entire	2–3 × 0.5–1, free apex ca. 1 mm, triangular-acute, entire	1.5–2.5 × 0.3–0.4, free apex ca. 0.5 mm, ovate-acute to acuminate, entire to eroded	3–4 × 1–1.5, free apex ca. 1.5 mm, ovate-acute to acuminate, entire	1–2 × 0.2–0.3, free apex ca. 0.4 mm, triangular-acute, entire
Stamen filament length (mm) and morphology	3–6, expanded ca. 0.5 mm in the lower half	6–8, expanded ca. 0.5 mm in the lower half	6–8, expanded ca. 0.5 mm in the lower half	8–11, expanded ca. 0.7 mm in the lower half	4–6, expanded ca. 0.2 mm in the lower half
Anther length (mm), shape and colour	0.5–0.7, oblong-ellipsoid, yellowish	0.8–1.2, ellipsoid, yellowish to reddish	0.4–0.6, ellipsoid, yellow	0.8–1.4, ellipsoid, yellowish	0.2–0.4, oblong-ellipsoid, yellowish
Ovules per placenta	4–6	6–10	5–6	7–12	12–20
Capsule size (mm)	2.5–3.5 × 0.6–2	2.5–3.5 × 1–2	2.5–3.5 × 1–2	3.5–5 × 1–2	2–3 × 0.5–1.0
Seed number and size (mm)	9–12, 0.7–1 × 0.3–0.5	12–22, 0.7–0.9 × 0.3–0.5	1–8, 0.7–1.1 × 0.3–0.4	12–22, 0.9–1.3 × 0.4–0.6	up to 45, 0.5–0.7× 0.2–0.3
Testa papillae length (μm)	12–33	3.5–9	10–43	55–110	4–17
Papillae morphology and distribution	heterogeneous, globose to conical-obtuse, denser on the distal part	subhomogeneous, conical-obtuse, sparse on the funicular part	heterogeneous, globose to conical-obtuse, denser on the distal part	subhomogeneous in shape, cylindrical-conical obtuse, dense all over the testa	homogeneous, conical-obtuse, denser on the distal part

## Data Availability

DNA sequence data generated in the present research are available at GenBank (https://www.ncbi.nlm.nih.gov/genbank/).

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
