# Peer review of "What Is Wrong with Frankenia nodiflora Lam. (Frankeniaceae)? New Insights into the South African Sea-Heaths"

_plants, 2023, doi:10.3390/plants12142630_

Round 1

Reviewer 1 Report

This paper sets out to demonstrate the existence of cryptic species within the genus Frankenia from southern Africa. The authors utilise morphology, aired with DNA sequence data (only nuclear ITS data) and geographic distribution and ecology to recognise two new species from the more arid areas of South Arica. This, together with a careful appraisal of type and other material and some nomenclatural detective work makes for a convincing paper and an interesting contribution to the flora of this region.

I have no major issues with this paper, and it is an interesting piece of research. However, it is not well written in places, and the English writing needs to be carefully checked. Some parts seem overly lengthy, and tables for example could be used to greater effect (details below). Also, I am not convinced that the title is appropriate. It certainly is rather long.

Also, why no chloroplast DNA data set? It would be most valuable to have this maternally inherited source of information to compare the ITS phylogeny to.

Specific comments:

Lines 81-87. The introduction of SEM of seeds is very abrupt and there is no motivation that precedes this to explain why seed micro-morphology may be of any taxonomic value. This particular approach and source of data needs to be introduced and explained.

Still on this topic, when the results of the SEM study are presented and discussed, there is no word of caution indicating that their observations are based entirely on seeds from a single specimen, and thus the authors have no clear idea of the extent of variation between specimens and within species in the features they observe. I am not suggesting the data be discarded, but the interpretation needs to be contextualised in terms of these limitations.

111-117 – this sentence / content needs to be reworded as it is a very long sentence.

Figure 2: Would it be possible to show the additive tree rather than an ultrametric one? A tree with branch lengths would be perhaps more informative about genetic distances between terminals and clades.

259-260: Correct the English

276-278. I think the name “Capeverdean” shoud be “Cape Verdean”

279: “…falls sister to…” should be “… is placed as sister to…”

281: “further check” – maybe “further investigated”?

352-366: This paragraph is badly written and needs careful English language editing.

Also in this paragraph, the authors state how scarce F. nodiflora is, and indicate a threat from alien invasive specie, but they then “chicken out” of providing a meaningful IUCN Red List assessment and opt for Data Deficient. Yes, there is not all the data needed for a full IUCN assessment, but the Extent of Occurrence and possibly Area of Occurrence can be assessed and used to guide the to a suitable criterion that will immediately put this species on the conservation radar.

472: What are intermittent saline riverbeds? Do you mean intermittently saline…?

479: What does “occurs concretely..” mean?

600: Similarly, use of the word “waadi” refers to a feature not associated with southern Africa, and once again there is the use of “intermittent riverbeds”… I don’t know what you are trying to say, but “intermittent” is the wrong word.

The discussion (656-717 especially): This is overly long and difficult to follow in the parts where the features of different species are being compared and discussed. Can this information not be tabulated for ease of comparison?

What I am missing from the discussion is a more nuanced analysis of the distribution patterns of these closely relates species – there are other studies on species and species complexes from this region, some of which relate speciation to edaphic and climatological factors. A brief comparison to these other studies may be relevant and interesting.

I find the conclusion to be brief and not entirely appropriate – it seems more like a Summary I don’t understand why the key is an appendix, seeing as the discussion went into great detail about differences between taxa. Surely including the key in the main body will be most informative. I don’t know if it is a journal requirement to have a conclusion, but it seems to be unnecessary for this kind of paper; the discussion (suitably reworked) followed by the key would be an appropriate way to end the paper.

Acknowledgements: Please check spelling of the Curator of PRE – as far as I know it is Erich van Wyk.

Some sections are well written, but others are in need of careful editing for English use.

Author Response

Dear Reviewer 1,

Many thanks for your detailed comments and suggestions that greatly improved the new version. I have responded to all your questionsw (as numbered points) in the attached Word file.

All the best.

Reviewer 2 Report

This manuscript is worth publishing with a few changes (precisions on the grid systems used). That's unfortunated that the main taxa discussed in the taxonomic part (i.e. F. nodiflora) was not sampled for phylogenetic analysis - it would have provided far more support to you. In addition, the description of phylogenetic analysis contains ambiguities (Boostrap computed - implicitly - for BI analyses, Boostrap values not provided from NJ tree - and in fact nothing presented from NJ analyses).

These comments are in the attached file

Author Response

(The authors gave the same response as above.)
